# The boundary-spanning mechanisms of Nobel Prize winning papers

Yakub Sebastian[1]*, Chaomei Chen[2]

**1** College of Engineering, IT & Environment, Charles Darwin University, Casuarina, Northern Territory, Australia, **2** College of Computing & Informatics, Drexel University, Philadelphia, PA, United States of America

* yakub.sebastian@cdu.edu.au

**Data Availability Statement:** The dataset used for this study can be accessed from the following DOI: https://doi.org/10.6084/m9.figshare.14633535.v1. Anyone who is interested to replicate our study can also fully reproduce the dataset based on the method detailed in this paper with 1) Dimensions

## Abstract

The breakthrough potentials of research papers can be explained by their boundary-spanning qualities. Here, for the first time, we apply the structural variation analysis (SVA) model and its affiliated metrics to investigate the extent to which such qualities characterize a group of Nobel Prize winning papers. We find that these papers share remarkable boundary-spanning traits, marked by exceptional abilities to connect disparate and topically-diverse clusters of research papers. Further, their publications exert structural variations on a scale that significantly alters the betweenness centrality distributions in existing intellectual space. Overall, SVA not only provides a set of leading indicators for describing future Nobel Prize winning papers, but also broadens our understanding of similar prize-winning properties that may have been overlooked among other regular publications.

## Introduction

### Background

The prospect of discovering methods and metrics for explaining and predicting breakthrough scientific papers, such as Nobel Prize winning papers, continues to fuel some of the most consequential studies into the science of science research [1–12]. On one hand, predicting significant scientific achievements is a challenging enterprise as an increasing number of studies have revealed the unpredictable nature of scientific success [13]. On the other hand, recent advances in digital libraries and scientific publication indexing services such as Dimensions, Lens.org, and Microsoft Academic Graph (in addition to the Web of Science and Scopus) provide us with an unprecedented level of digital access to hundreds of millions of scientific publications and metadata [14, 15]. These resources present new opportunities for developing the next-generation computational methods for explaining and predicting breakthroughs in science [16].

In this paper, we propose structural variation analysis as a promising computational model that explains a major underlying mechanism of the intellectual impacts of Nobel Prize winning papers, especially in terms of their boundary-spanning qualities. Our method sheds light on the distinctive quality of these papers from the standpoint of Structural Variation Theory [4].

API and 2) CiteSpace, both of which are accessible at no cost. Alternatively, the dataset can be manually curated without the Dimensions API via https://app.dimensions.ai/discover/publication by any users at no cost. Such curated datasets can also be processed by CiteSpace.

**Funding:** The author(s) received no specific funding for this work.

**Competing interests:** The authors have declared that no competing interests exist.

We premise this work on the assumption that connecting otherwise disparate clusters of knowledge is a key mechanism behind transformative scientific discoveries, which also shares the vision of literature-based discovery research pioneered by Swanson [17]. Such approach is advantageous as it renders scientific impact measurable, explainable, and actionable through a rich set of structural variation metrics computed from large bibliographic networks, as will be demonstrated in this paper.

## Motivation

Nobel Prize is considered by many as the epitome of scientific achievements [18, 19]. The towering prestige of the Nobel Prize within the scientific community and the extraordinary esteem conferred to their laureates motivated a number of important studies into the mechanisms behind winning the award [1, 20, 21]. But measuring and explaining the scientific impact of Nobel Prize winning papers is not easy [3, 22–25]. There are limitations in solely relying on citation counts, journal impact factors, or other summative metrics as they do not provide clear theoretical and mechanistic explanations on what makes certain scientific breakthroughs more important than the others.

The Structural Variation Theory, in contrast, provides a useful theoretical framework that explains Nobel Prize winning papers by their shared boundary-spanning mechanisms. There is a growing body of evidence that support this conceptual view. Nobel laureates possess the exceptional intellectual agility that allowed them to transcend own specialities and appropriate fresh concepts from other disciplines that enable intellectual quantum leaps [26]. [6] found that the propensity of a paper to be highly cited was correlated with its ability to make highly novel combinations of prior knowledge. And recently [19] suggested that researchers should include 'a balanced mixture of new and established elements' as a key ingredient in making a successful scientific career.

The theory is operationalized as the Structural Variation Analysis (SVA) module in Cite-Space, which is a popular application for visualizing and analyzing trends and patterns in scientific literature [5, 27] (available at https://citespace.podia.com/). A number of SVA metrics have been proposed to measure different dimensions of a paper's breakthrough properties. However, their value as potentially leading indicators of future Nobel Prize winning papers have not been systematically studied [5, 28–30]. [4] previously studied two cases of Nobel Prize in Physiology or Medicine in their introduction of Structural Variation Theory, i.e. the discovery of the bacterium *Helicobacter pylori* and its role in peptic ulcer disease [31] and the discovery of principles for introducing specific gene modification in mice by the use of embryonic stem cells [32]. But their investigation was limited to measuring the structural and temporal properties of the winning papers. Note that these properties are *lagging* indicators of success because they can only be calculated from the citation patterns of a paper, which may take many years to accumulate. In contrast, it is highly desirable to have a new set of *leading* indicators that allow the breakthrough potentials of a paper to be measured immediately upon its publication.

There is also a unique challenge associated with consolidating the impact metrics of a Nobel Prize. The award is often shared by more than one laureates, suggesting the presence of several related breakthrough papers. Because these papers may be published many years or decades between each other [11, 33], they add an additional layer of complexity when measuring the breakthrough characteristics for a single Nobel Prize.

## Contributions

Two research problems are addressed in this paper:

1. Do Nobel Prize winning papers exert exceptional boundary-spanning mechanisms upon their underlying intellectual structures as postulated by the Structural Variation Theory? If yes, what SVA metrics best signals the boundary-spanning mechanisms of these papers and why?

2. What strategy should be used to consolidate the breakthrough characteristics when multiple winning papers are associated with a Nobel Prize?

In answering these questions we make the following research contributions. First, within the constraint of our selected case studies, we demonstrate that Nobel Prize winning papers exhibit highly distinguishable boundary-spanning properties. We empirically show two SVA metrics, Centrality Divergence and Entropy, as strongly salient measures of the structural variations exerted by these papers. Our results provide fresh insights into mechanisms that may be at work behind transformative research. For cases where there are multiple winning papers, we propose synthesizing an artificial pseudopaper as a way to consolidate their structural variation potentials. Finally, we also provide some practical advice when using CiteSpace's SVA module in search of promising breakthrough papers.

The rest of this paper is organized as follows. We first review some related work, followed by descriptions of our selected Nobel Prize cases. Then, we explain in detail the SVA method, metrics, and experimental results. We conclude this paper with some discussions on the significance of our findings, their limitations, and possible directions for future research in this area.

## Related work

### The mechanisms of scientific breakthroughs

There is a rich tapestry of prior work that aim to provide mechanistic explanations for scientific breakthroughs [34, 35]. Here, we cast our work against more recent developments in this research, especially those that concern Nobel Prize papers. [6] found that the probability of a research paper to be highly cited ('high impact' or 'hit' papers) doubled if it made highly novel combinations of prior knowledge apart from strengthening well-established, conventional associations. Although their finding supports the correlation between boundary-spanning properties of a paper and its future high impact, it offers no specific insights for papers associated with the Nobel Prizes. [36] investigated the relationships between prior awards and the citation count of landmark papers of the Nobel Prize in Physiology or Medicine. The study, which covered prizes awarded between 1983 and 2012, found that citation ranking and impact factor did not always positively correlate with the chance of winning Nobel Prizes. In a similar vein [7], also studied how the combinations of prior work can be correlated to the future impact of a paper as measured by its citation count. They found that among papers indexed in the Web of Science and in the U.S. Patent Office database, high impact papers were more likely to include those that cited younger references. The work, however, did not specifically account for the mechanisms behind Prize-winning papers.

Another group of works incorporated a diverse set of success indicators to predict future Nobel Prize winners. [21] proposed a combination of total citation counts and the $h$-index as a new citation index measure. Using this measure, they found that Nobel laureates tend to author fewer but highly cited papers, applicable to at least five Nobel laureates in physics. For the Nobel Prize in Economics [37], discovered that winning the John Bates Clark Medal was a predictive factor for winning the Nobel prize. [38] traced the career patterns of all Nobel Prize laureates from 1900 to 2016 and found that although significant alterations in the co-authorship structure and dramatic changes in the post-Nobel research directions were unique to most laureates, nothing was remarkable about their pre-Nobel career trajectories in

comparison to those of non-laureates. [39] conducted a vast bibliometric analysis of five hundred most cited and influential chemists and physicists indexed in the Web of Science from 1900 to 2006. These included the nominees and laureates of Nobel Prize in physics and chemistry. Their results suggested significant difficulties in explaining the publications of Nobel Prize winners when relying on measures such as citation-based ranking and Freeman's degree centrality. [40] used LDA topic modelling and SVM classifier to predict when breakthrough will occur. A novel feature, *innovation score*, was introduced to measure how far ahead or behind time are the topics contained in a particular paper. They found the correlations between high citation counts and high innovation scores. The study is limited to papers published in the WWW and SIGIR conference proceedings.

### Network properties and structural variations

Certain properties of bibliographic network surrounding a paper may offer valuable clues to their scientific potentials. For instance [20], modelled the research networks of the laureates of Nobel Prize in Physiology or Medicine between 1969 and 2011. Levels of productivity, impact, co-authorship and collaborative patterns were then calculated from these networks. Consistent with [21], they found that the laureates produced fewer but highly cited papers. The authors also examined the average degree, density, modularity, and communities of co-authorship networks to find that Nobel laureates demonstrated distinctive abilities for performing scientific brokering roles that close existing structural holes within the networks. This is in line with the central premise of the structural variation theory.

More recently [41], explored the ability of six network centrality indices in capturing the process of attributing 'success' to a scientific work by human experts. Their scenarios included emulating expert opinions on the impact of nearly half a million physics papers published by the American Physical Society from year 1893 to 2009, where successful papers were identified as those that have won a Nobel Prize (48 papers in total). The results indicated that, for a direct citation network, the PageRank scores best captured expert opinions concerning high-impact qualities of a scientific paper. Note that building a direct citation network requires that papers receive prior citations and recognition by other papers. In [8] the authors constructed a large scientific prize network that also included Nobel laureates. Analysis done on the network revealed the interlocking of small, elite subdisciplinary areas formed by previously awarded prizes. By regressing the number of scientific prizes won by each winner on a set of variables, the authors found that the propensity of winning scientific prizes could be explained by a person's university prestige, high *h*-index, and by the fact that they belonged to a prize-winning genealogical network. In a separate but related study [42], investigated if papers associated with Nobel Prizes in physics, medicine or chemistry between 1995 and 2017 were heavily clustered only in a handful disciplines. They found that only 5 out 114 fields of science accounted for more than half of the studied Nobel Prizes. The winning papers originated from clusters with high-citation density, although no variable was suggested as being predictive of these winners.

There are several other works in this area. [43] proposed a search algorithm that measures four characteristics of breakthrough papers (including several Nobel Prize winning papers): application-oriented research impact, cross-disciplinary research impact, researchers-inflow impact, and discoverers-intra-group impact. Similar to [12], computing these measures requires that a paper's emerging citation patterns be available. [9] developed methods for detecting potential breakthrough papers that depend on first categorising every paper into one of four citation-based typologies: lowly cited publications, moderately cited publications, highly cited publications, and outstanding publications.

Our work is most related to the work by Min et al. [12]. The authors computed a paper's scientific breakthrough potentials by measuring the structural variations observable from the network of its citing papers. Their proposed metrics include average clustering coefficient, average degree, maximum closeness centrality, maximum eigenvector centrality, and number of connected components in the network. For evaluation, 116 Nobel Prize winning papers in all fields were collected with techniques introduced by [44, 45]. By comparing these against a group of control papers (i.e. regular papers with comparable citation counts), they found that Nobel Prize winning papers had a higher number of connected network components, indicative of some forms of boundary-spanning mechanisms.

The method proposed by Min et al. above significantly differs from ours in that their structural variations were computed from the network structure of a Nobel Prize winning papers' *citing papers*. For this to work, they required that the first layer of citing papers had already accumulated over time following the publication of the Nobel paper concerned. As such, their proposed metrics are *lagging* indicators of scientific breakthrough. In contrast, we are proposing a set of SVA metrics as *leading* indicators of breakthrough that can be computed at the time of a paper's publication. From a theoretical point of view, our SVA method is derived directly from Chen's structural variation theory and therefore has a clear theoretical underpinning. In contrast, we feel that the theoretical motivation behind Min et al.'s selected structural variation metrics was somewhat unclear. We also differ in terms of dataset quality, where we based the current selection of Nobel Prize winning papers on a dataset systematically curated and published in [33]. This encourages transparency and reproducibility of results.

More fundamentally, our approach makes it possible to measure the transformative potentials as soon as a scholarly article is published because all the necessary information is readily accessible upon its publication. Furthermore, our approach could explain the underlying generic mechanism regarding how creative ideas can be constructed across multiple disciplines. A practical implication of this explanatory power is significant because the identified mechanisms, as demonstrated in this paper, suggest to researchers how they can better recognize other similar mechanisms in action and apply them in their own field of research. There are also other benefits, including increased immunity against the Matthew Effect and improved capability to capture scientific merit of previously neglected scholarly works [46, 47]. Altogether, these distinctive characteristics of our approach clearly separate our work not only from Min et al. but also the rest of citation-based predictive models. We discuss these benefits in more detailed in the Discussion section.

### Previous applications of SVA

SVA has previously been used in various scientometric investigations. [5] studied the statistical correlation between early SVA metrics (i.e. modularity change rate, cluster linkage, and centrality divergence) with the likelihood of a paper to become highly cited. The SVA model was also applied to identify important papers in the field of regenerative medicine [48]. [29] used the modularity change rate as the basis for calculating four types of scholarly impact of individual researchers and most recently [30] employed SVA for analysing the transformative potentials of COVID-19 literature.

### Case studies

In this work, we focus on the latest collection of Nobel Prize winning papers that were systematically curated by [33] (downloadable from doi.org/10.7910/DVN/6NJ5RN). The dataset covers key publications of 92.4% of all Nobel laureates from 1900 to 2016. We limit this study to 3 cases of the Nobel Prize in Physiology or Medicine, even though our method is consistently

applicable to other Nobel Prize categories. The selected cases are Nobel Prizes winners in 2012, 2014, and 2016. The first two cases (2012, 2014) were selected on the basis that their landmark papers are the youngest in Li et al.'s dataset [33]. We reason that younger papers provide richer bibliographic data for analysis from decades worth of baseline literature. The third case (2016) was selected for its unique characteristics, to be explained shortly.

### Case 1: Nobel Prize in Physiology or Medicine 2012

John B. Gurdon and Shinya Yamanaka were jointly awarded the Nobel Prize in Physiology or Medicine 2012 for the discovery that mature cells can be reprogrammed to be pluripotent stem cells [49]. The discovery occurred in two phases. First, in a landmark experiment [50] successfully transplanted mature, fully differentiated intestinal cell nuclei into the egg cell of a frog. The egg cell then developed into a normal tadpole, demonstrating for the first time the reversibility of cell specialisation process, which is the cornerstone for future cell reprogramming techniques.

More than 40 years later, in 2006, Shinya Yamanaka and colleague discovered a combination of four genes that can be used to induce pluripotent stem cells (or iPSCs) from mature cells called fibroblasts in laboratory environments [51, 52]. It was a major scientific breakthrough given the fact that iPSCs can be prepared from mature human cells and used for developing all kinds of body cells. The field of regenerative medicine was born. Together, the works of Gurdon and Yamanaka not only give new insights into the development of cells and organisms, but also provide scientists with formidable tools for creating radically new forms of medical treatments.

Four papers were selected as directly associated with this prize [33]. The selection was made by inspecting bibliographic information from the Nobel Prize official website, the laureates' Nobel Prize lectures and online CVs, and by applying a few other criteria. Two of the papers are attributed to Gurdon [50, 53], while the remaining to Yamanaka and his colleagues [51, 52]. All papers are highly cited in recognition of their scientific contributions. At the time when we collected their bibliographic data [53], was cited 321 times [50], 701 times [51], 16,123 times, and [52] 12,866 times.

### Case 2: Nobel Prize in Physiology or Medicine 2014

The Nobel Prize in Physiology or Medicine 2014 is shared among John O'Keefe, May-Britt Moser and Edvard I. Moser for their discoveries of cells responsible for the positioning system in the brain [54]. Their works answer the long mystery of how the brain maps its surrounding space and navigates through complex environments. Similar to Case 1, the scientific breakthrough occurred in two phases. First, O'Keefe discovered the first component of brain positioning system called the 'place cells' [55]. These are specialised nerve cells responsible for forming spatial maps. Subsequently in 2005 May-Britt and Edvard Moser discovered the 'grid cells', which is the second crucial component in the brain's positioning system [56]. Grid cells are responsible for creating a coordinate system, which is crucial for precise positioning and pathfinding. The discoveries of both types of cells opens a new way for understanding how an ensemble of brain cells perform higher cognitive functions that include thinking, memory, and planning. Two breakthrough papers [55, 56], are attributed to this prize [33].

### Case 3: Nobel Prize in Physiology or Medicine 2016

Yoshinori Ohsumi received the Nobel Prize in Physiology or Medicine 2016 for his discovery of the mechanisms for autophagy [57]. Autophagy is the process by which cells degrade and recycle their own cellular components. It is a key cellular process that helps eliminate

intracellular viruses and bacteria. Because damages in the autophagy machinery have been associated with multiple diseases, new drugs can be designed to fix the autophagy process. Prior to Ohsumi's groundbreaking work, little was known as to how autophagy takes place. There are two winning papers for this prize. In the first, Ohsumi proved the existence of autophagy in yeast cells by introducing a novel cell engineering technique that disrupts the degradation process in the yeast cells [58]. The disruption led to the accumulation of autophagosomes in the cell vacuole, which was observable under a microscope. The autophagosomes provide the direct evidence for autophagy process in cells. In the second paper, Ohsumi further postulated that genes instrumental to autophagy must have been activated when the autophagosomes accumulated in the cell vacuole [59]. To find out, he exposed the yeast cells a specific chemical that allowed fifteen autophagy-specific genes to be isolated. This succession of findings complete his extraordinary discovery.

As mentioned above, the breakthrough papers for this case possess peculiar characteristics that make it an interesting case study. Both Ohsumi's papers have been noted as exemplary of *under-cited influential* landmark papers [22]. This means that the paper should been more frequently cited that they actually are and there is little explanation as to why and how such papers could satisfy the conventional breakthrough criteria. Therefore, our aim is to provide an alternative explanation for their intellectual impact using SVA.

## Materials and methods

### Structural Variation Analysis (SVA)

The theoretical foundation and central premise of the structural variation theory are well-documented and will not be further elaborated in this paper [4, 5]. SVA, which is currently available as a module in the CiteSpace software [27], operationalizes structural variation theory by providing tools and metrics for measuring the intellectual impact of a paper in terms of the structural changes it exerted on its underlying co-citation network.

A co-citation network or graph comprises of a set of interlinked nodes, where a node represents a cited reference (i.e. a scientific document that has been cited by another scientific document in any given citation or bibliographic dataset). A pair of nodes are connected by a link if they have been cited together by at least one other scientific document. By scientific document, we generally mean research papers, patents, or any other citable information artefact. On the assumption that topically-related research papers have a greater tendency to share similar cited references than those which are not topically related, a co-citation network provides a powerful representation for identifying groups of research papers that represent a disciplinary or sub-disciplinary areas of research. The intuition and theoretical foundation behind document co-citation are well elaborated in [60].

SVA then applies the Louvain community detection algorithm [61] on a given co-citation network to identify such document groupings as described above. These groupings correspond to the idea of 'communities' in community detection literature, such that there are denser links among members of the same community than members from different communities. For the remainder of this manuscript, we refer to these groups or communities of research papers as 'clusters'.

Specifically, SVA focuses on detecting boundary-spanning mechanisms in a co-citation network of scientific papers, such that papers capable of establishing a higher number of new links between previously disconnected clusters in a co-citation network are considered to be more scientifically impactful than the others. Note that the detection of other types of discovery mechanisms is beyond SVA's design scope.

Detecting SVA signals of a paper depends on two factors: (1) the SVA metrics used and (2) the SVA network modelling. Prior studies indicated that different SVA metrics have varying strength of correlation with the eventual impact of a paper [5]. Importantly, their performance as leading indicators for Nobel Prize winning papers have never been studied before. The network from which SVA metrics are calculated must be configured in ways that could amplify SVA signals. For example, widening the temporal coverage of SVA baseline network can bring up richer structures in the network. In the following sections we describe the SVA metrics in mathematical terms and also recommend a promising network modelling strategy.

## SVA metrics

We study seven SVA metrics: (a) modularity change rate, (b) cluster linkage, (c) centrality divergence, (d) harmonic mean of (a)-(c), (e) within-cluster link, (f) between-cluster link, and (g) entropy. Here the mathematical properties of the metrics are briefly explained. Readers should refer to [5, 30] for more details. Table 1 lists common notations and definitions used throughout our metric formulations.

**Modularity change rate ($\Delta M$), cluster linkage ($CL$), and centrality divergence ($C_{KL}$).** These are the most established SVA metrics [5]. The $\Delta M$ of paper $a$ measures the relative structural change of a network as a result of additional information added by paper $a$ onto its baseline network. $\Delta M$ is mathematically defined as follows [5, 29]:

$$\Delta M(a) = \frac{Q(G_s, C) - Q(G_a, C))}{Q(G_s, C)} \tag{1}$$

The higher the $\Delta M$ score, the greater the extent of relative structural changes exerted by paper $a$ upon its baseline network. To obtain $C$ and $Q$, the Louvain community detection algorithm is used [61].

The Cluster Linkage ($CL$) metric measures the overall structural change exercised upon a network by paper $a$ in terms of novel connections added between clusters. $CL$ ignores within-cluster links. Eq 2 explains how the *Linkage* score of a network $G$ with partition $C$ is calculated, where $n$ is the total number of nodes in $G$, $\lambda_{ij}$ is an edge function between nodes $v_i$ and $v_j$, and

**Table 1. Basic notations and definitions.**

| | |
|---|---|
| $G(V, E)$ | Network $G$ is a document co-citation network; $v \in V$ are nodes representing cited references; $e \in E$ are links between nodes that are cited together at least once. |
| Paper $a$ | Paper that is the target of SVA. |
| $\tau_a$ | The year paper $a$ is published. |
| $Y_{(t, a)}$ | $t$-year(s) of publication immediately prior to $\tau_a$. |
| $G_s$ | Baseline network $G$ generated from a set of articles $s$ published in the period $Y_{(t, a)}$. |
| $G_a$ | $G_s$ plus novel co-citation links introduced by paper $a$. |
| $Q(G, C)$ | Overall modularity of $G$ obtained by an arbitrary partitioning $C$. |
| $K$ | Total number of clusters in a network. |
| $C_B(v, G)$ | Betweenness centrality of node $v$ in network $G$. |

$\epsilon_{ij}$ is an adjustable weight of the between-cluster link between both nodes.

$$Linkage(G, C) = \frac{\sum_{i \neq j}^{n} \lambda_{ij} \epsilon_{ij}}{K}$$

$$\lambda_{ij} = \begin{cases} 0, v_i \in c_j \\ \\ 1, v_i \notin c_j \end{cases}$$

(2)

The *CL* score of paper *a* is then calculated as a weighted ratio between the increase or decrease in Linkage score if a network owing to novel between-cluster links added by *a* in $G_a$, in comparison to the original Linkage score of $G_s$ before the links are added. This is shown in Eq 3. *CR* denotes the number of cited references in paper *a* that contributed to making novel between-cluster links, whereas *NR* is the total number of references it cited. This means that, all other things being equal, the $\frac{CR}{NR}$ weighting scheme allows *CL* to reward a paper with a high percentage of boundary-spanning references.

$$CL(a) = \left( \frac{Linkage(G_a, C) - Linkage(G_s, C)}{Linkage(G_s, C)} \right) \times 100 \times \left( \frac{CR}{NR} \right)$$

(3)

The third metric, Centrality Divergence ($C_{KL}$), captures the structural variations induced by paper *a* in terms of the divergence of the distribution of betweenness centrality of nodes in the baseline network. Unlike $\Delta M$ and *CL*, $C_{KL}$ does not depend on the partitioning of a network and is formulated as follows:

$$C_{KL}(G_s, a) = \sum_{i=0}^{n} p_i \cdot \log\left( \frac{p_i}{q_i} \right)$$

(4)

where $p_i = C_B(v_i, G_s)$ and $q_i = C_B(v_i, G_a)$. The degree of structural changes measured by $C_{KL}$ is defined in terms of the Kullback-Leibler Divergence [5].

Previous research have suggested that different SVA metrics are sensitive to different kinds of global structural variations. For predicting future citation counts of papers in four research domains (terrorism, mass extinction, complex network analysis, and knowledge domain visualization) [5] found that on overall *CL* was the best predictor. He also found $C_{KL}$ to be particularly useful for capturing boundary-spanning characteristics of a paper at interdisciplinary levels. On the other hand, $\Delta M$ was useful for characterising the evolution of individual authors' scholarly impacts over time from a structural variation theory point of view [29].

**Within-cluster link ($\alpha$) and between-cluster link ($\beta$).** [28] introduced within-cluster and between-cluster links as additional measures of structural variation. The within-cluster link ($\alpha$) metric calculates the rate of *novel* within-cluster links added by paper *a* divided by the number of all within-cluster links:

$$\alpha(a) = \frac{\text{no. of novel within-cluster links introduced by } a}{\text{no. of all within-cluster links}}$$

(5)

The between-cluster link ($\beta$) metric is calculated in the same way as $\alpha$, except that it only focuses on novel between-cluster links that connect nodes from disparate clusters. The $\beta$ score of paper *a* is computed as follows:

$$\beta(a) = \frac{\text{no. of novel between-cluster links introduced by } a}{\text{no. of all between-cluster links}}$$

(6)

**Entropy ($E$).** A citing paper's entropy ($E$) is calculated as a standard information entropy of the proportion of its cited references over the clusters of the underlying co-citation network. Entropy is calculated below:

$$E(a) = -\sum_{i=1}^{n} p(a,i) \cdot \log(p(a,i)) \tag{7}$$

where $p(a,i) = \frac{\text{no. of references in } a \text{ that belong to cluster } i}{\text{no. of references in } a}$ and $n$ is the total number of clusters in a merged network. Given its mathematical definition, a citing paper with the maximum entropy would have its cited references evenly distributed across all clusters. In contrast, a paper with the least entropy (0) would have all its references residing in a single cluster.

**Harmonic ($H$).** The harmonic score of paper $a$ ($H(a)$) is the harmonic mean of three structural variation metric scores, i.e. $\Delta M(a)$, $CL(a)$, and $C_{KL}(a)$ [30]. It summarizes the impact of a paper from three different aspects of structural variations. We calculate $H(a)$ as follows:

$$H(a) = \frac{3 \cdot \Delta M(a) \cdot CL(a) \cdot C_{KL}(a)}{\Delta M(a) \cdot CL(a) + \Delta M(a) \cdot C_{KL}(a) + CL(a) \cdot C_{KL}(a)} \tag{8}$$

## Network modelling

Beside the choice of SVA metrics, another factor that significantly influences the outcomes of structural variation analysis is the network configurations. This section explain our approach. We follow the standard practice of SVA analysis that recommends document co-citation network as the primary network representation [5, 29, 30]. There are two non-trivial network modelling components to be addressed here: (1) the method for acquiring relevant literature that forms the basis for constructing a co-citation network, and (2) finding suitable network configurations.

**Cascading citation expansion.** Adequately identifying the most representative body of scholarly publications determines the quality of the resultant network and its subsequent SVA analyses. For this purpose, we used the Cascading Citation Expansion (CCE) method proposed by [62], which offers a flexible method to improve the quality of data retrieved for systematic scientometric reviews. Our selected Nobel Prize winning papers served as the initial seed of CCE expansion.

The CCE procedures are as follows. Bibliographic records associated with the three cases of Nobel Prize awards were downloaded from Dimensions [63] in July 2020 using the CCE function in CiteSpace. The CCE process started with a set of seed publications (i.e. the previously identified Nobel Prize winning papers) and retrieved publications cited by the seed publications as a step backward expansion, as well as publications that cited the seed publications as a one step forward expansion. In practice, users may specify multiple steps of expansion in each direction to meet their needs. Refer to [62] for more details of the CCE expansion process.

To obtain the dataset used in this study, we applied a 1-step backward and 1-step forward expansion from the seed publications identified previously (refer to the case studies section above). For each seed paper, its DOI was used as a a start. If the DOI is missing, then we would use the Dimensions publication ID, such as pub.1076143041. Below is a list of all seed papers with their corresponding DOI and Microsoft Academic Graph ID (MAGID) numbers.

1. Case 1, Seed 1 ($S_{1,1}$): [53]. Times cited: 321.
   DOI 10.1016/0012-1606(62)90043-X. MAGID 2094753906.

2. Case 1, Seed 2 ($S_{1,2}$): [50]. Times cited: 701.
   PubmedID 13951335. MAGID 2153824299.

3. Case 1, Seed 3 ($S_{1,3}$): [51]. Times cited: 16,123.
   DOI 10.1016/j.cell.2006.07.024. MAGID 2125987139.

4. Case 1, Seed 4 ($S_{1,4}$): [52]. Times cited: 12,866.
   DOI 10.1016/j.cell.2007.11.019. MAGID 2138977668.

5. Case 2, Seed 1 ($S_{2,1}$): [55]. Times cited: 3,385.
   DOI 10.1016/0006-8993(71)90358-1. MAGID 2052515926.

6. Case 2, Seed 2 ($S_{2,2}$): [56]. Times cited: 2,050.
   DOI 10.1038/nature03721. MAGID 1970792572.

7. Case 3, Seed 1 ($S_{3,1}$): [58]. Times cited: 787.
   10.1083/jcb.119.2.301. MAGID 2126801593.

8. Case 3, Seed 2 ($S_{3,2}$): [59]. Times cited: 1,135.
   10.1016/0014-5793(93)80398-E. MAGID 2011580247.

Table 2 shows the basic bibliographic profiles of each dataset retrieved with the Cascading Citation Expansion (CCE) technique. The column *Unique Records* denotes the combined non-duplicate bibliographic records retrieved using CCE for each Nobel Prize case, seeded with previously identified seed publications. The percentages indicate the amount of valid DOIs and References in the records. At the time of retrieval, abstracts were not yet available from Dimensions.

**Network parameters.** The retrieved datasets above were then used to construct document co-citation networks. These network must be configured to improve the ability to capture SVA signals [5]. One may *a priori* assume that larger networks are more likely to include novel co-citation links. However, indefinitely increasing the network sizes could be counter-productive. A higher level of noise can be introduced into the increasingly larger network. Also, the performance of metrics such as $\Delta M$ may be more influenced by the specific locations the novel links are added, instead of by the size of the network [5]. Finally, from a practical standpoint, constraining the network size has the benefit of keeping reasonable SVA running time in CiteSpace.

We explain the key parameters for configuring co-citation networks for SVA. First, the basic parameters were applied to all current cases. These are **Link Retaining Factor** (*LRF*), **Maximum Links Per Node** (*Max Links*), **Look Back Years** (*LBY*), and parameter *e*. We set LRF to 3, which retains the number of strongest links up to three times the number of nodes in a network. *Max Links* was set to 10, ensuring that at most ten strongest links are retained by a node. Both parameters have been empirically shown to increase the clarity of a network's visualization. *LBY* and *e* were set to -1 (i..e unlimited look back years) and 0.00, respectively. *LBY* affects the number of cited references to be included for each citing paper, whereas *e* controls the number of top cited references appearing in the network. Both settings ensure that these parameters are free from subjective preferences of the analyst.

We focus our present study on the following parameters:

**Table 2. Bibliographic profiles of datasets retrieved using CCE method for the selected Nobel Prize cases.**

| Case | Unique Records | Range | DOI | References | Date of Retrieval |
|------|----------------|-------|-----|------------|-------------------|
| 1 | 45,628 | 1952–2020 | 99.40% | 99.19% | 7/16/2020 |
| 2 | 14,885 | 1948–2020 | 99.70% | 99.68% | 7/16/2020 |
| 3 | 16,330 | 1951–2018 | 97.97% | 96.50% | 7/13/2020 |

1. **Scaling factor** ($k$). Parameter $k$ modifies the $g$-index of the network. The $g$-index is the largest number that equals the average number of citations of the most highly cited $g$ publications [64]. Setting higher $k$ values produces larger networks, vice versa. We set this initially to 5.

2. **Publication time frame**. This parameter controls the time frame of an SVA analysis relative to a target paper. It corresponds to the notation $Y_{t,a}$ in the Method section. For all case studies, we set publication time frame to 5 years prior to the publication year of a Nobel Prize winning paper. This means that if the paper was published in 2007 then the publication time frame spans from 2002 to 2007.

3. **Sliding window**. This parameter dictates the number of years required to build the baseline network, which is necessary for computing most of our SVA metrics. This parameter was set to 5 years for all cases to maintain consistency. This implies that if the seed paper was published in 2007, the baseline network for calculating its SVA metrics will be constructed from all papers published five years prior to 2007 (i.e. from 2002 to 2006).

## Artificial breakthrough papers

We mentioned at the introduction to this paper that one of our key research contributions is a method for consolidating the structural variation effects of multiple seed papers associated with the same Nobel Prize. For this, we propose artificially creating a pseudopaper that encapsulates all novel links made by the associated seed papers. Given two seed papers $s_1$ and $s_2$ published in $\tau_{s_1}$ and a later year $\tau_{s_2}$, respectively, we propose the following algorithm:

1. Create a pseudopaper $Ps(s_1 \oplus s_2)$ to replace both seed papers.

2. Let $Ps(s_1 \oplus s_2)$ cite all references cited by $s_1$ and $s_2$, with duplicated references removed.

3. Remove $s_1$ and $s_2$ from the co-citation network in year $\tau_{s_1}$ and $\tau_{s_2}$, respectively.

4. Place $Ps(s_1 \oplus s_2)$ in the co-citation network of year $\tau_{s_2}$. The pseudopaper should not be placed in year $\tau_{s_1}$ because it may cite references that only appear in $\tau_{s_2}$.

5. Remove without replacement any citation to $s_1$ in year $\tau_{s_1}$, and replace any to citation to either $s_1$ or $s_2$ in year $\tau_{s_2}$ with a citation to $Ps(s_1 \oplus s_2)$.

6. Run SVA.

## Results

This section presents the results of our SVA analyses on the selected cases of Nobel Prize in Physiology or Medicine winners. In these results, we will refer to the target Nobel Prize winning papers as seed papers (denoted by notation $S_{x,y}$). Recall that these were used for seeding the Cascading Citation Expansion process described previously. For example, paper $S_{1,1}$ denotes the first seed paper identified for Case 1 [53], $S_{1,2}$ the second seed paper for Case 1 [50], $S_{2,1}$ the first seed paper for Case 2, and so on. Any other paper is the non-Nobel Prize winning paper. Refer to the previous section for a complete list of all the seed papers.

### SVA on Case 1

The co-citation network for $S_{1,1}$ and $S_{1,2}$ (Gurdon's papers) was constructed from the publications between 1957 and 1962. The dataset, obtained with CCE method, included 51 papers and

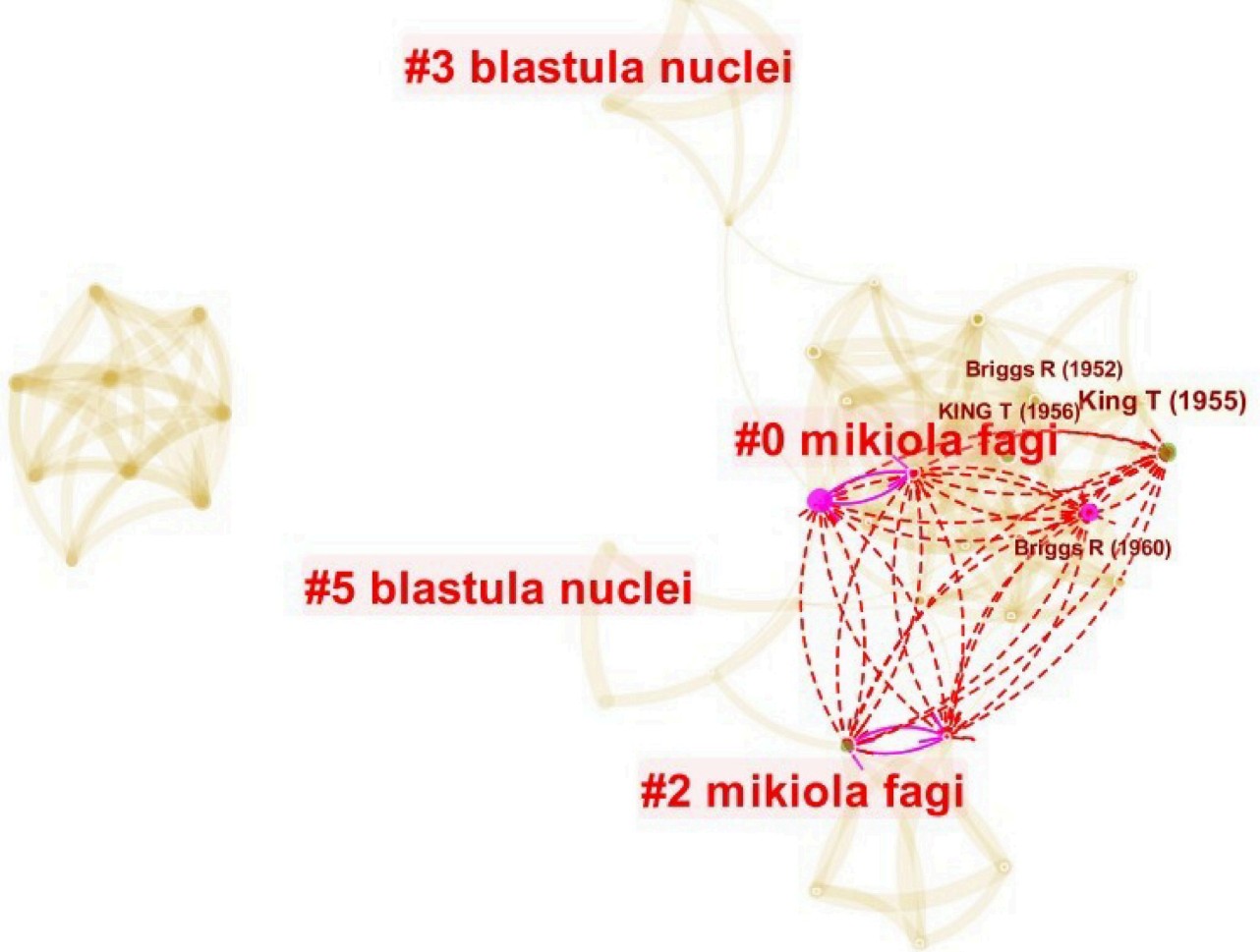

CiteSpace, v. 5.7.R4 (64-bit)
January 30, 2021 2:29:53 PM ACST
WoS: C:\Users\ysebastian\.citespace\SVA_A1\data
Timespan: 1957-1962 (Slice Length=1)
Selection Criteria: g-index (k=5), LRF=3.0, LBY=-1, e=0.0
Network: N=47, E=157 (Density=0.1452)
Largest CC: 32 (68%)
Nodes Labeled: 1.0%
Pruning: None
Modularity Q=0.2944
Weighted Mean Silhouette S=0.9399
Harmonic Mean(Q, S)=0.4483

#3 blastula nuclei

Briggs R (1952)

KING T (1956) King T (1955)

#0 mikiola fagi

Briggs R (1960)

#5 blastula nuclei

#2 mikiola fagi

**Fig 1. Case 1, Seed 1 ($S_{1,1}$).** The boundary-spanning mechanism exhibited by Gurdon's landmark publication in 1962.

786 unique cited references. Fig 1 visualizes the co-citation network. The numbered labels stand for various clusters detected from the network. The red dash lines represent novel co-citation links introduced by $S_{1,1}$. One could observe in the network that Gurdon's first landmark paper $S_{1,1}$ drew many new connections that span across various parts of the network's core largest connected component. His second paper, $S_{1,2}$, did not appear in the network, suggesting that it did not add novel links. This is supported by the fact that nearly forty-percent of its references were already covered by $S_{1,1}$. As a result, there were also not many important structural holes left in the network to be bridged by $S_{1,2}$.

**Table 3. Case 1.** The SVA scores of top-6 most cited papers in the network (1957-1962). SVA score ranges: $\Delta M$:0.0–64.69, **CL**:-15.00–145.71, $C_{KL}$:0.0–0.58, **H**:0.0–1.72, $\alpha$:0.0–0.5, $\beta$:0.0–1.0, **E**: 0.0–1.01. The subscript next to each SVA score of the seed paper indicates the paper's relative rank according to that metric, sorted in a descending order.

| Citation | $\Delta M$ | CL | $C_{KL}$ | H | $\alpha$ | $\beta$ | E | DOI |
|---|---|---|---|---|---|---|---|---|
| 321 ($S_{1,1}$) | $64.69_1$ | $145.71_1$ | $0.58_1$ | $1.72_1$ | $0.5_1$ | $1_*$ | $1.01_1$ | 10.1016/0012-1606(62)90043-x |
| 68 | 0 | -3.43 | 0.14 | 0 | 0 | 1 | 0.69 | 10.1016/0012-1606(62)90006-4 |
| 59 | 0 | -15 | 0.15 | 0 | 0 | 1 | 0.69 | 10.1016/0012-1606(62)90004-0 |
| 45 | 21.28 | 11.54 | 0.22 | 0.67 | 0 | 1 | 0.5 | 10.1016/0012-1606(62)90037-4 |
| 43 | 0 | 0 | 0 | 0 | 0 | 0 | 0 | 10.1007/bf00577042 |
| 29 | 29.87 | 2.86 | 0.17 | 0.54 | 0 | 1 | 0.56 | 10.1002/jcp.1030600404 |

$S_{1,1}$ denotes the seed paper [53] and the asterisk * represents a tie with other papers.

This initial SVA outcome is highly encouraging. Table 3 shows the SVA scores of seed $S_{1,1}$ compared with other top cited papers in the result set. In this relatively small network (47 nodes, 157 links), only 6 papers made novel co-citation links. $S_{1,1}$ topped nearly all SVA metrics (except $\beta$ where it performed equally with the other papers), clearly indicating strong impacts exerted by its publication on the underlying intellectual structure. Its high entropy score suggests that the paper cited a highly diverse set of references in various clusters. Its $CL$ score greatly stands out, which also indicates that $S_{1,1}$ introduced many novel co-citation links that connected disparate clusters. Finally, the seed paper also emerged at the top of the $H$ metric, suggesting its prominence along the most established SVA metrics ($\Delta M$, $CL$, $C_{KL}$).

Four decades passed between Gurdon's landmark papers and Yamanaka's groundbreaking works, during which the intellectual structures would have changed considerably. Consequently, SVA was run on $S_{1,3}$ and $S_{1,4}$ independently from from $S_{1,1}$ and $S_{1,2}$ to precisely demonstrate the structural changes effected by Yamanaka's discovery against the state-of-the-art in the field. The publication analysis time frame spans from 2001 to 2007, calculated as five years prior to the publication the earlier seed ($S_{1,3}$). The merged network consisted of 454 nodes and 2,373 links, with 2,126 papers making novel co-citation links. The network is visualized in Fig 2 and the SVA results are given in Table 4. Both seed papers are the most cited papers in the result set. For comparison, we plotted novel co-citation links made by the third most cited paper in the result, namely the paper by [65]. The purple lines indicate the existing (non-novel) links found in the paper, whereas the red dash lines represent newly introduced novel links. A further examination on this competing paper revealed that it is in a close competition to Yamanaka's work as the authors also successfully isolated four protein transcription factors for reprogramming human somatic cells to pluripotent stem cells in the same year as Yamanaka. Then, in Fig 3 we show novel links introduced by $S_{1,3}$ and $S_{1,4}$. It is easy to see that, in contrast to [65] who contributed relatively few novel links, Yamanaka's papers added a much higher number of novel links that span across the far reaches of the network. Comparing the two later figures also suggests that the structural variations exerted by $S_{1,3}$ are likely to be more significant that that of $S_{1,4}$ not only because the former added a lot more novel links than the latter but also because the links span the boundaries of multiple clusters in the network.

Consistent with the visualizations, the results in Table 4 suggest that $S_{1,3}$ performed better than $S_{1,4}$ as it ranked first by metric E, second in $C_{KL}$ and H, and third according to $CL$. Its high entropy score accounts for the highly diverse set of clusters that it connected (see again Fig 3). In contrast, $S_{1,4}$ did not feature prominently by any SVA standard. Overall, however, the structural variations induced by Yamanaka's breakthrough papers appear less striking than that of Gurdon's paper. There could be several possible explanations for these. First, due to the

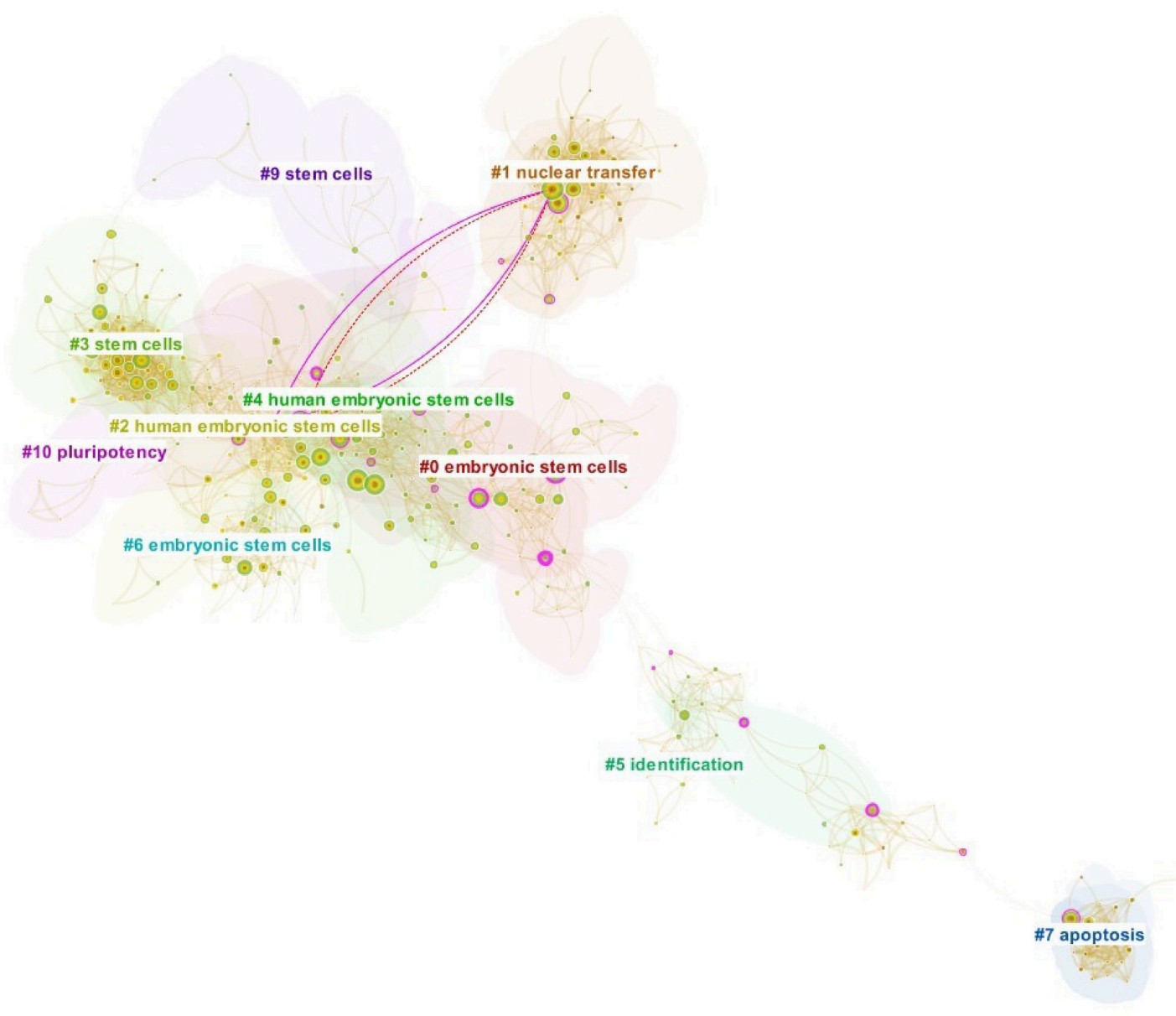

**Fig 2. Case 1, Seed 3 ($S_{1,3}$) and Seed 4 ($S_{1,4}$).** The co-citation network surrounding the publication of $S_{1,3}$ and $S_{1,4}$ (2001–2007). The purple and red dash lines are co-citation links made by one of the Yamanaka's competitors [65]. The numbered cluster labels were generated using CiteSpace's implementation of Latent Semantic Indexing.

intense competition among regenerative cell biologists and the rapid proliferation of publications in this research discipline, there were perhaps fewer significant structural holes to span over. Second, it is plausible that the breakthrough components are so distributed among Yamanaka's two papers that SVA signals can only marginally be detected from each paper when observed independently. Therefore, a method that consolidates these breakthrough elements into a single paper may overcome this limitation. Akin to Gurdon's case, it is may be that the poor SVA signal from $S_{1,4}$ can be explained by the fact that some of its otherwise novel links have already been made in $S_{1,3}$. We have observed that at least 30% references in $S_{1,4}$ had already been cited by $S_{1,3}$.

**Table 4. Case 1.** The SVA scores of top-10 most cited papers in the network (2001-2007). SVA score ranges: The ranges of the obtained SVA metric scores are: **ΔM**:0.0–89.36, **CL**:-76.24–494.69, $C_{KL}$:0.0–0.87, **H**:0.0–2.54, $\alpha$:0.0–1.0, $\beta$:0.0–1.0, **E**: 0.0–2.10. The subscript next to an SVA score of each seed paper indicates the paper's relative rank according to that metric, sorted in a descending order.

| Citation | ΔM | CL | $C_{KL}$ | H | $\alpha$ | $\beta$ | E | DOI |
|---|---|---|---|---|---|---|---|---|
| 16123 ($S_{1,3}$) | $65.89_{36}$ | $267.53_3$ | $0.82_2$ | $2.42_2$ | $0.62_{91}$ | $0.99_{591}$ | $2.1_1$ | 10.1016/j.cell.2006.07.024 |
| 12866 ($S_{1,4}$) | $4.18_{548}$ | $-51.89_{1596}$ | $0.01_{487}$ | $0.03_{319}$ | $0.53_{347}$ | $0.77_{831}$ | $0.95_{356}$ | 10.1016/j.cell.2007.11.019 |
| 7257 | 0 | -21.4 | 0 | 0 | 0 | 0.33 | 1.1 | 10.1126/science.1151526 |
| 3836 | 0 | -8 | 0 | 0 | 0.49 | 0 | 0 | 10.1038/nature05874 |
| 3702 | 8.25 | -7.01 | 0.35 | 0.96 | 0.25 | 1 | 0.67 | 10.1016/j.cell.2006.02.041 |
| 3309 | 0 | -42.86 | 0 | 0 | 0.52 | 0 | 0 | 10.1038/nature05934 |
| 2132 | 7.96 | -20.48 | 0.02 | 0.05 | 0.29 | 0.93 | 0.96 | 10.1038/nature05944 |
| 1844 | 0 | 0 | 0 | 0 | 0 | 0 | 0 | 10.1126/science.1141319 |
| 1829 | 0 | -5.71 | 0 | 0 | 0 | 1 | 0.69 | 10.1002/dvg.20335 |
| 1781 | 25.92 | -7.77 | 0.35 | 0.98 | 0.2 | 1 | 1 | 10.1038/ng1760 |

## SVA on Case 2

Similar to the first case, in this Nobel Prize case a significant number of years lapsed between seed papers $S_{2,1}$ [55] and $S_{2,2}$ [56]. Hence, we ran SVA independently on each seed paper. The results for $S_{2,1}$ in Table 5 indicate that the paper wrought remarkable structural changes upon its baseline network as it topped all SVA metrics, except for *CL* and $\beta$ which ranked it at fifteenth and third places, respectively. The underlying network consisted of 89 nodes and 399 links, with a total of 38 citing papers made novel co-citation links. Fig 4 visualizes the novel links introduced by $S_{2,1}$ to the network (1966–1971). They span over the boundaries of Cluster #0 and Cluster #2. The identical cluster labels indicate the close affinity of both clusters' research topics. This result provides a clear evidence of boundary-spanning mechanisms at work behind O'Keefe's landmark paper.

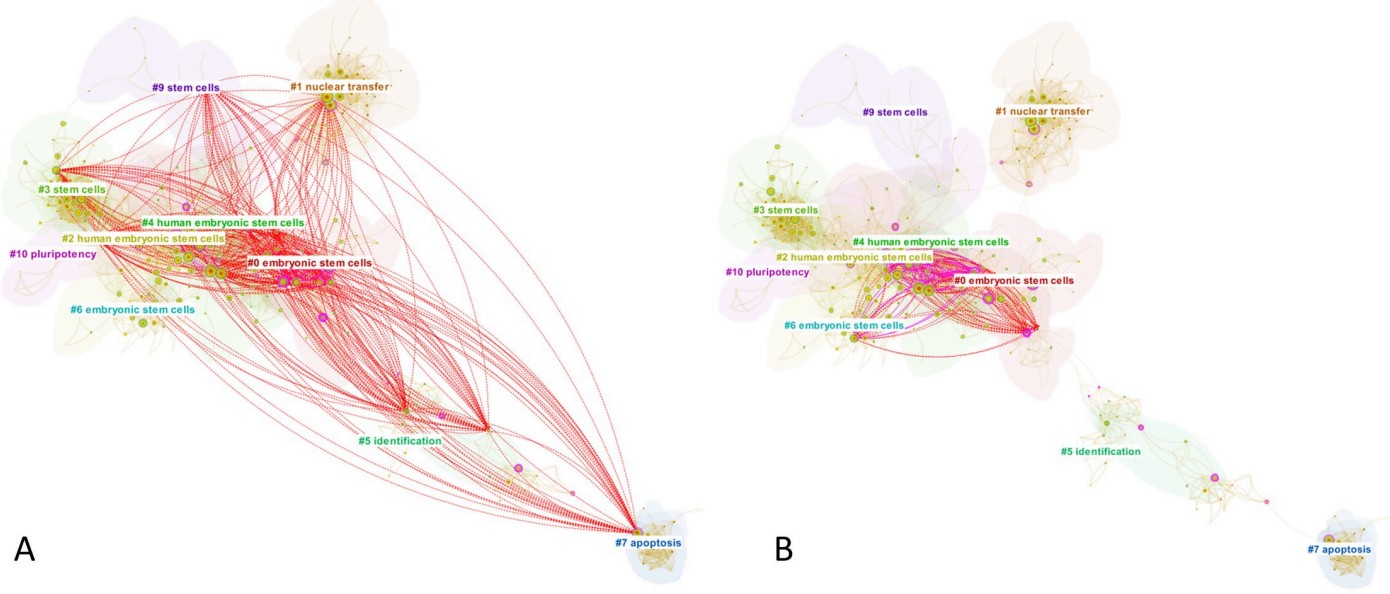

**Fig 3. Novel co-citation links introduced by $S_{1,3}$ and $S_{1,4}$.** **(A)** depicts novel links introduced by $S_{1,3}$ [51], and **(B)** shows novel links added by $S_{1,4}$ [52]. Note that the underlying network is identical to that is shown in Fig 2.

**Table 5. Case 2.** The SVA scores of top-10 most cited papers in the network (1966-1971). SVA score ranges: **ΔM**:0.0–75.41, **CL**:-48.05–0.00, $C_{KL}$:0.0–0.54, **H**:0.0–1.58, $\alpha$:0.0–1.0, $\beta$:0.0–1.0, **E**: 0.0–1.3. The subscript next to an SVA score of each seed paper indicates the paper's relative rank according to that metric, sorted in a descending order.

| Citation | ΔM | CL | $C_{KL}$ | H | $\alpha$ | $\beta$ | E | DOI |
|---|---|---|---|---|---|---|---|---|
| 3385 ($S_{2,1}$) | $75.41_1$ | $-48.05_{15}$ | $0.54_1$ | $1.58_1$ | $1_1$ | $0.89_3$ | $1.33_1$ | 10.1016/0006-8993(71)90358-1 |
| 356 | 3.7 | -1.61 | 0.02 | 0.06 | 0.21 | 0.52 | 0.61 | 10.1007/bf00142518 |
| 294 | 0 | 0 | 0 | 0 | 0 | 0 | 0.69 | 10.1016/0028-3932(71)90005-4 |
| 291 | 0 | 0 | 0 | 0 | 0 | 0 | 0.69 | 10.1016/0006-8993(71)90303-9 |
| 200 | 0 | -17.86 | 0.01 | 0 | 0.4 | 0 | 0 | 10.1007/bf00234246 |
| 195 | 0 | 0 | 0 | 0 | 0 | 0 | 0 | 10.1113/jphysiol.1971.sp009681 |
| 154 | 0 | -25 | 0 | 0 | 0.5 | 0 | 0 | 10.1113/jphysiol.1971.sp009508 |
| 135 | 0 | -11.34 | 0.02 | 0 | 0 | 1 | 0.69 | 10.1016/0031-9384(71)90172-7 |
| 132 | 0 | 0 | 0 | 0 | 0 | 0 | 0 | 10.1016/0031-9384(71)90235-6 |
| 131 | 0 | 0 | 0 | 0 | 0 | 0 | 0 | 10.1016/0042-6989(71)90005-8 |

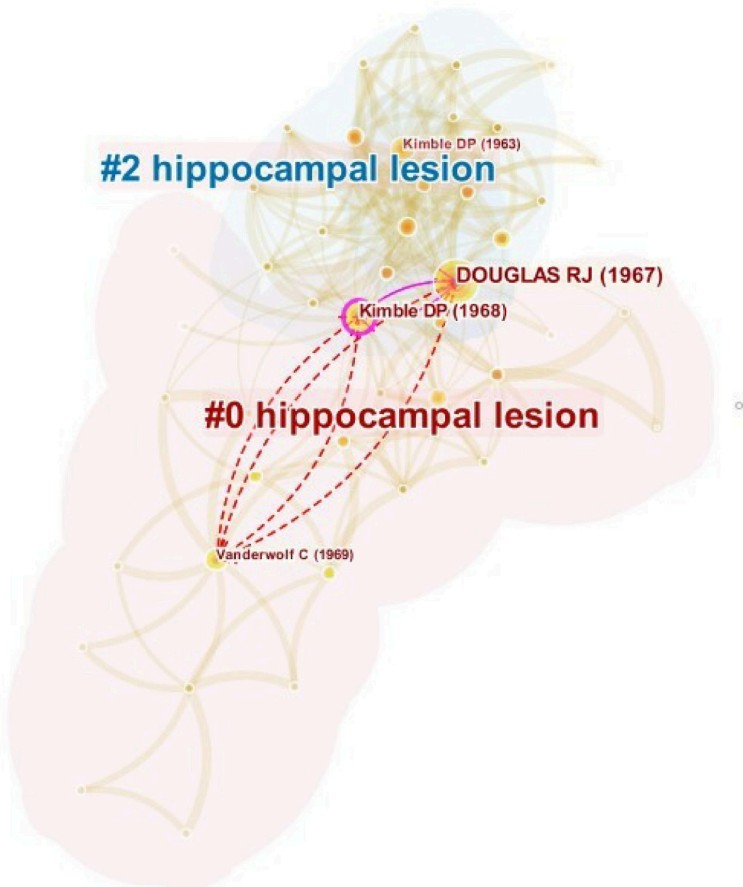

**Fig 4. Case 2, Seed 1 ($S_{2,1}$).** The boundary-spanning mechanism exhibited by O'Keefe's landmark publication in 1971.

**Table 6. Case 2.** The SVA scores of top-10 citing papers in the network (2000-2005). SVA score ranges: **ΔM**:0.0–62.54, **CL**:-93.62–54.84, $C_{KL}$:0.0–0.25, **H**:0.0–0.67, $\alpha$:0.0–1.0, $\beta$:0.0–1.0, **E**: 0.0–1.43. The subscript next to an SVA score of each seed paper indicates the paper's relative rank according to that metric, sorted in a descending order.

| Citation | ΔM | CL | $C_{KL}$ | H | $\alpha$ | $\beta$ | E | DOI |
|---|---|---|---|---|---|---|---|---|
| 2050 ($S_{2,2}$) | $0_{48}$ | $36.42_4$ | $0.25_1$ | $0_{42}$ | $0.44_{84}$ | $0.89_{127}$ | $1.32_8$ | 10.1038/nature03721 |
| 1181 | 0 | -2.89 | 0 | 0 | 0 | 1 | 0.64 | 10.1016/j.neuron.2005.05.002 |
| 1174 | 0 | -4.3 | 0 | 0 | 0.45 | 0.6 | 0.6 | 10.1038/nrn1607 |
| 940 | 62.54 | -13.82 | 0.01 | 0.02 | 1 | 1 | 1.04 | 10.1038/nature03687 |
| 680 | 0 | 0 | 0 | 0 | 0 | 0 | 0.56 | 10.1152/jn.00697.2004 |
| 673 | 0 | -5.85 | 0.03 | 0 | 0.53 | 1 | 0.41 | 10.1113/jphysiol.2004.078915 |
| 649 | 0 | -10.07 | 0.02 | 0 | 0.67 | 1 | 0.5 | 10.1016/j.neuron.2005.02.028 |
| 637 | 0 | -8.93 | 0.01 | 0 | 0 | 1 | 0.64 | 10.1371/journal.pbio.0030402 |
| 584 | 0 | -4.8 | 0.07 | 0 | 0.59 | 0.92 | 0.76 | 10.1111/j.1469-7580.2005.00421.x |
| 502 | 0 | 93.62 | 0.14 | 0 | 0.47 | 0.9 | 1.16 | 10.1002/hipo.20113 |

The SVA result for $S_{2,2}$, i.e. May-Britt and Edvard Moser's discovery paper [56] is presented in Table 6. Similar to Yamanaka's papers, there is less clear-cut evidence for its structural variations from the metric point of view. Despite being the most cited paper in the result set, $S_{2,2}$ ranked no. 1 only in $C_{KL}$ and fourth by $CL$. $H$ was not useful due to zero $\Delta M$ and this time $E$ did not give a particularly strong signal. Instead, one may observed in this result that [66] (DOI:10.1038/nature03687) scored highly for $\Delta M$ where it ranked no. 1. The paper successfully discovered that neurons are activated differently by extremely different pictures of people, places, or objects, itself an important discovery. This may suggest another great potential of SVA as a discovery tool. It can be used to find other impactful papers that should have received a better recognition from the scientific community.

The underlying network for $S_{2,2}$ are shown in part **(A)** of Fig 5, which covered 197 nodes and 1,137 links. There were 277 citing papers that made novel co-citation links. There are six major clusters in the network that overlapped with each other. The novel links added by $S_{2,2}$

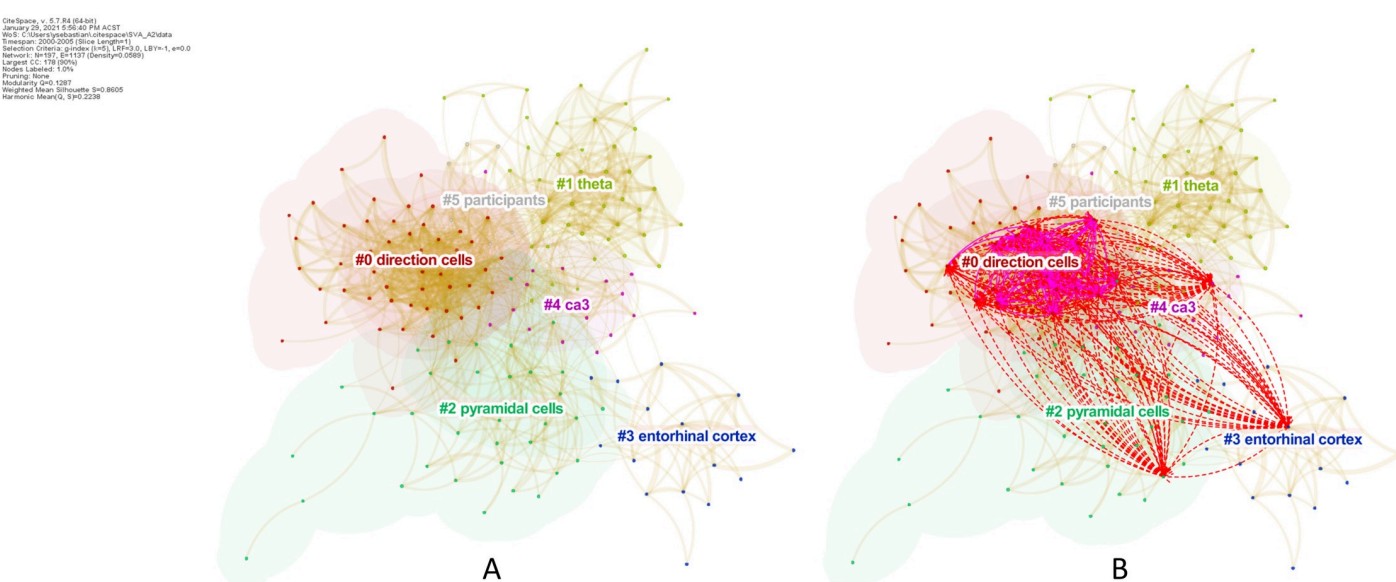

**Fig 5. Case 2, Seed 2 ($S_{2,2}$).** The co-citation network surrounding $S_{2,2}$ (2000–2005). The clusters were labeled with cited publications' keywords. **(A)** shows the underlying network. **(B)** illustrates the novel links introduced by $S_{2,2}$.

are illustrated in Fig 5 part **(B)**. The highly dense links cover 5 out of the 6 major clusters. The existing links (purple links) are concentrated on the largest cluster (Cluster #0), which is the center of the network, where most research works were traditionally concentrated. Importantly, we can see that the novel links (red dash lines) significantly extended the boundaries of the existing links to include pyramidal cells and entorhinal cortex research topics, which are located at the extremities of the network.

## SVA on Case 3

The two winning papers contributing to this award [58, 59] were published by Ohsumi and his colleagues at nearly the same time. Hence, we covered both papers in a single SVA analysis. Table 7 shows that neither $S_{3,1}$ nor $S_{3,2}$ ranked highly by any SVA metric. In addition, both papers were not the most cited among their cohort, making our result consistent with a previous study that considered Ohsumi's papers as the so-called *under-cited influential* papers [22]. This atypical characteristic poses a unique challenge for SVA that we will attempt to solve presently.

There are 225 nodes and 1,225 links in the merged network, as shown in Fig 6. There were 305 citing papers that made novel co-citation links. It is interesting to find that although the boundary-spanning property of $S_{3,1}$ is obvious from the visualization (i.e the paper connected two opposing clusters on the far sides of the network), the paper's SVA scores were relatively unremarkable compared to other papers. Fine-tuning the network parameters may be necessary to better uncover its structural variations. Alternatively, the boundary-spanning mechanisms of both seed papers may need to be further consolidated. Altogether, this demonstrates the non-trivial challenge in extrapolating a uniform SVA configuration across different scenarios.

For Ohsumi's second landmark paper ($S_{3,2}$), the network visualization in Fig 7 part **(A)** shows the relatively fewer novel links it added. Fig 7 part **(B)** allows us to visually compare $S_{3,1}$ and $S_{3,2}$ against the most highly cited paper by Sollner et al. [67] found in the current result. One can tell that, unlike Ohsumi, Sollner et al. did not make a significant number of new connections despite being more highly cited. This is evident from the fewer red dash lines compare to the purple ones. This highlights the discrepancy between a conventional measure of impact such as total citation count and what could be considered as groundbreaking from the standpoint of the Nobel Prize committee. Here SVA provides a promising alternative that better explains the breakthrough qualities of a paper.

**Table 7. Case 3.** The SVA scores of the top-10 citing papers in the network (1987-1993). SVA score ranges: **ΔM**:0.0–80.62, **CL**:0.0–49.42, $C_{KL}$:0.0–0.27, **H**:0.0–0.70, $\alpha$:0.0–1.0, $\beta$:0.0–1.0, **E**: 0.0–1.55. The subscript next to an SVA score of each seed paper indicates the paper's relative rank according to that metric, sorted in a descending order.

| Citation | ΔM | CL | $C_{KL}$ | H | $\alpha$ | $\beta$ | E | DOI |
|---|---|---|---|---|---|---|---|---|
| 2463 | 0 | -17.31 | 0 | 0 | 0.19 | 0 | 0 | 10.1038/362318a0 |
| 1482 | 0 | 0 | 0 | 0 | 0 | 0 | 0 | 10.1016/0092-8674(93)90376-2 |
| 1477 | 0 | -8.54 | 0 | 0 | 0.57 | 0 | 0 | 10.1083/jcb.116.5.1071 |
| 1135 ($S_{3,2}$) | $0_{220}$ | $-16.67_{197}$ | $0_{118}$ | $0_{125}$ | $1_{19}$ | $0_{183}$ | $0_{220}$ | 10.1016/0014-5793(93)80398-e |
| 787 ($S_{3,1}$) | $53.29_{17}$ | $-12.16_{168}$ | $0.03_{45}$ | $0.09_{16}$ | $0_{96}$ | $1_{12}$ | $1.04_{29}$ | 10.1083/jcb.119.2.301 |
| 769 | 0 | -11.77 | 0.02 | 0 | 0.47 | 0.71 | 0.34 | 10.1038/355409a0 |
| 669 | 0 | 0 | 0 | 0 | 0 | 0 | 0 | 10.1091/mbc.3.12.1389 |
| 614 | 0 | 0 | 0 | 0 | 0 | 0 | 0 | 10.1172/jci115849 |
| 569 | 0 | 0 | 0 | 0 | 0 | 0 | 0.69 | 10.1111/1523-1747.ep12359590 |
| 534 | 0 | -2.15 | 0.04 | 0 | 1 | 0 | 0 | 10.1002/j.1460-2075.1993.tb05813.x |

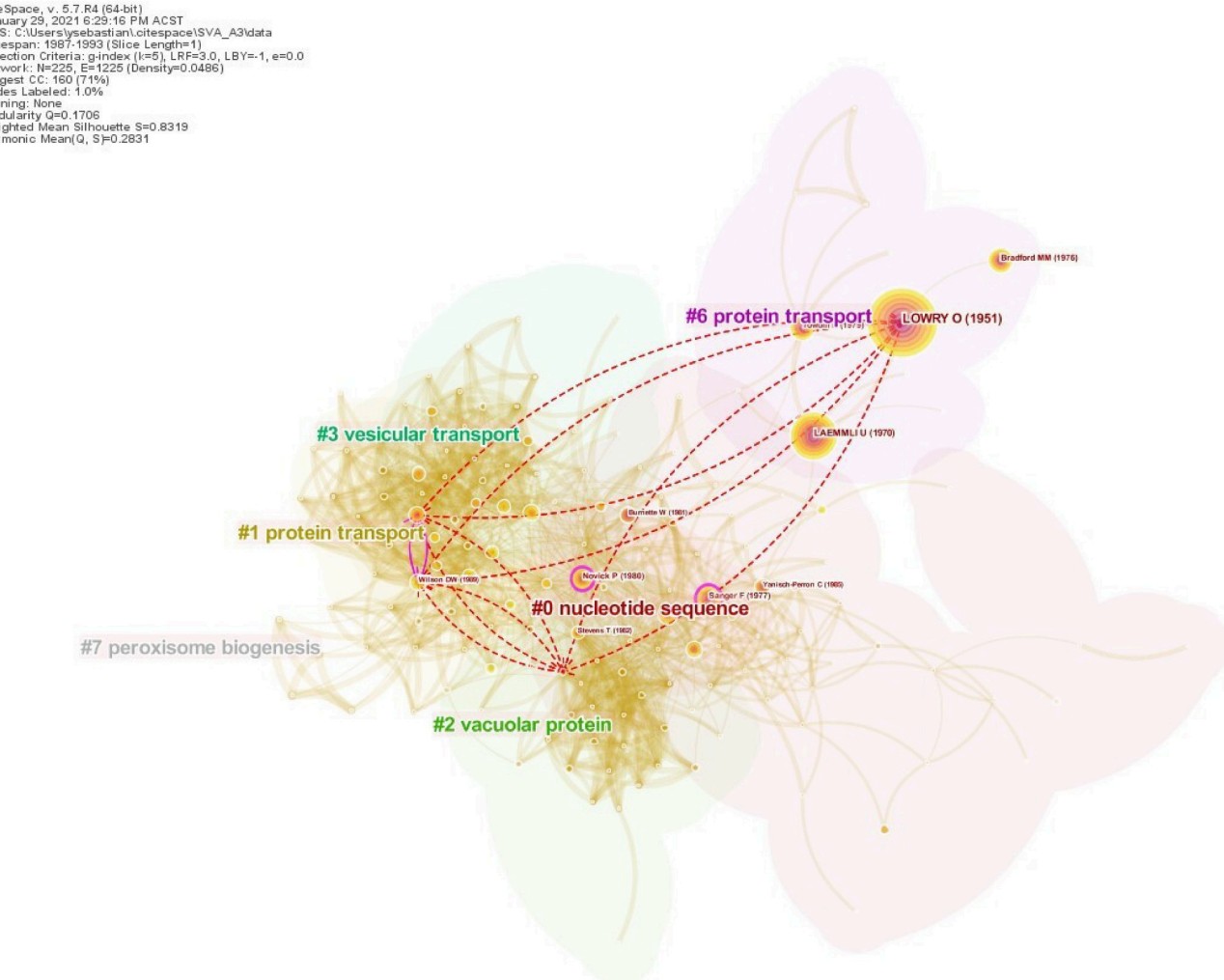

CiteSpace, v. 5.7.R4 (64-bit)
January 29, 2021 6:29:16 PM ACST
WoS: C:\Users\ysebastian\.citespace\SVA_A3\data
Timespan: 1987-1993 (Slice Length=1)
Selection Criteria: g-index (k=5), LRF=3.0, LBY=-1, e=0.0
Network: N=225, E=1225 (Density=0.0486)
Largest CC: 160 (71%)
Nodes Labeled: 1.0%
Pruning: None
Modularity Q=0.1706
Weighted Mean Silhouette S=0.8319
Harmonic Mean(Q, S)=0.2831

**Fig 6. Case 3, Seed 1 ($S_{3,1}$).** The co-citation network surrounding Ohsumi's breakthrough papers (1987–1993), showing novel links added by $S_{3,1}$ [58]. Cluster labels were generated with log-likelihood ratio and the node sizes correspond to the degree of betweenness centrality of a cited reference.

## SVA performance with artificial pseudopapers

We summarize the performance of all SVA metrics for the three Nobel Prize cases in Table 8. The marginal performances in independently detecting SVA signals of $S_{1,3}$ and $S_{1,4}$ (Case 1), $S_{2,2}$ (Case 2), and all seed papers in Case 3 prompted us to consider a different approach to measure structural variations. In this section, we present the results obtained from applying our proposed pseudopaper strategy. Still from Table 8 we can observe that consolidating the underperfoming seed papers into a pseudopaper dramatically improved many key SVA signals. For Case 1, $Ps(S_{1,3} \oplus S_{1,4})$ topped the $CL$, $C_{KL}$, $H$, and $E$. This improvement is significant considering that only $E$ could previously detect the structural variations induced by $S_{1,3}$. On the contrary, a pseudopaper did not greatly benefit Gurdon's publications. $S_{1,1}$ has already performed well in virtually all SVA metrics on its own accord. Adding $S_{1,2}$ made no difference as this second paper did not contribute novel links to the network as mentioned earlier.

A similar better performance was obtained for Case 2 as its pseudopaper $Ps(S_{2,2} \oplus \rho_{S_{2,2}})$ ranked first according to $CL$, $C_{KL}$, $H$, and $E$. Without the pseudopaper approach, $S_{2,2}$ only

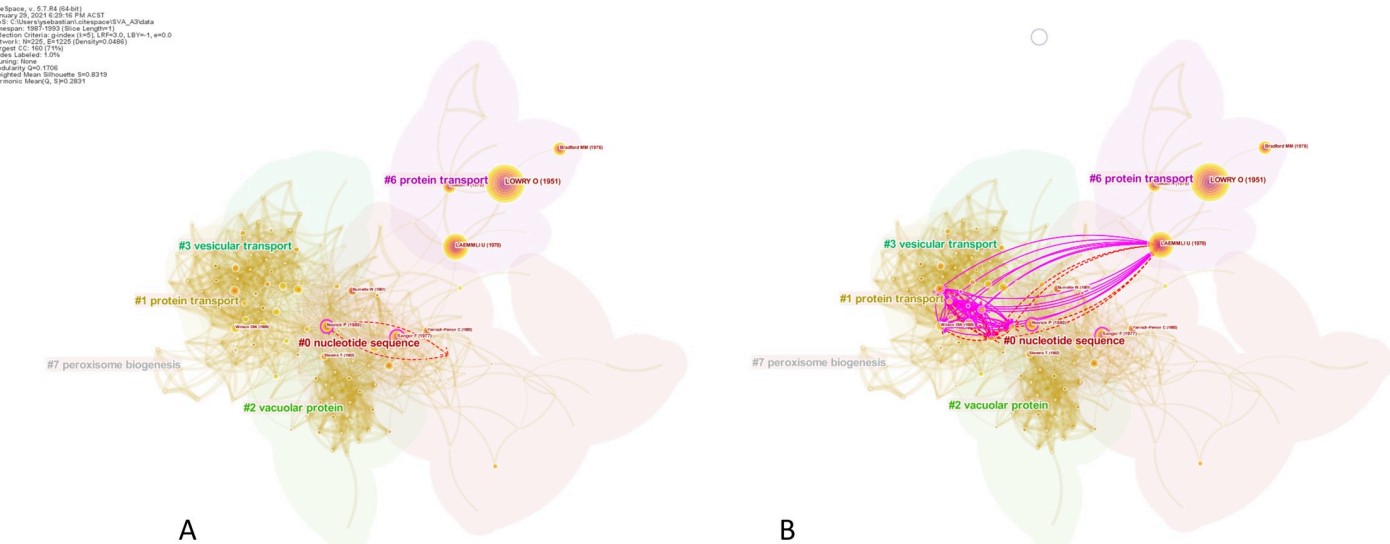

**Fig 7. Case 3, Seed 2 ($S_{3,2}$).** Comparing the structural variations induced by Ohsumi's papers against the most cited paper in our SVA result set [67] (see Table 7). **(A)** outlines the novel links induced by $S_{3,2}$ [59]. **(B)** compares the novel links induced by [67]. The underlying network is identical to that in Fig 6.

stood out by $C_{KL}$. The new notation $\rho_{2,2}$ denotes a regular, non-winning paper that was paired with seed paper $S_{2,2}$. We will explain the selection of $\rho_{2,2}$ shortly at the end of this section. It is important to note that in addition to improving the overall SVA detection results, a pseudopaper preserved the top SVA metrics of the underlying seed papers (i.e. $E$ in Case 1 and $C_{KL}$ in Case 2). We intentionally excluded O'Keefe;s paper ($S_{2,1}$) from the current pseudopaper analysis due to its previous good SVA performance.

Unfortunately for Case 3, strong SVA signals of seed papers remained elusive even after the application of the pseudopaper strategy (Table 8, second last row). So, as suggested previously, further fine-tuning SVA parameters such as the scaling factor $k$ may be the necessary next step in this case. Increasing $k$ from 5 to 25 successfully improved the SVA detection performance

**Table 8. Summary performance of SVA metrics for Cases 1–3.** The pseudopaper strategy amplifies SVA signals that are otherwise hard to detect from the original seed papers. $Ps(s_1 \oplus s_2)$ denotes a pseudopaper generated from seed papers $s_1$ and $s_2$. Where applied, $\rho_s$ represents a collection of non-seed paper(s) published by the Nobel laureates of $s$ in the same year as seed paper $s$'s publication.

| Seeds | Citation | $\Delta M$ | CL | $C_{KL}$ | H | $\alpha$ | $\beta$ | E |
|---|---|---|---|---|---|---|---|---|
| $S_{1,1}$ | $321_1$ | $64.69_1$ | $145.71_1$ | $0.58_1$ | $1.72_1$ | $0.5_1$ | $1_*$ | $1.01_1$ |
| $S_{1,2}$ | $701_{n/a}$ | n/a | n/a | n/a | n/a | n/a | n/a | n/a |
| $Ps(S_{1,1} \oplus S_{1,2})$ | $1022_1$ | $64.69_1$ | $68.52_1$ | $0.54_1$ | $1.63_1$ | $0.25_1$ | $1_*$ | $1.01_1$ |
| $S_{1,3}$ | $16123_1$ | $65.89_{36}$ | $267.53_3$ | $0.82_2$ | $2.42_2$ | $0.6_{291}$ | $0.99_{591}$ | $2.1_1$ |
| $S_{1,4}$ | $2866_2$ | $4.18_{548}$ | $-51.89_{1596}$ | $0.01_{487}$ | $0.03_{319}$ | $0.53_{347}$ | $0.77_{831}$ | $0.95_{356}$ |
| $Ps(S_{1,3} \oplus S_{1,4})$ | $28989_1$ | $57.19_{53}$ | $807.52_1$ | $1.21_1$ | $3.57_1$ | $0.72_{113}$ | $0.99_{513}$ | $2.02_1$ |
| $S_{2,1}$ | $3385_1$ | $75.41_1$ | $-48.05_{15}$ | $0.54_1$ | $1.58_1$ | $1_1$ | $0.89_3$ | $1.33_1$ |
| $S_{2,2}$ | $2050_1$ | $0_{48}$ | $36.42_4$ | $0.25_1$ | $0_{42}$ | $0.44_{84}$ | $0.89_{127}$ | $1.32_8$ |
| $Ps(S_{2,2} \oplus \rho_{S_{2,2}})$ | $2167_1$ | $2.34_{37}$ | $1361.17_1$ | $0.42_1$ | $1.08_1$ | $0.45_{82}$ | $0.9_{121}$ | $1.45_1$ |
| $S_{3,1}$ | $787_5$ | $53.29_{17}$ | $-12.16_{168}$ | $0.03_{45}$ | $0.09_{16}$ | $0_{96}$ | $1_{12}$ | $1.04_{29}$ |
| $S_{3,2}$ | $1135_4$ | $0_{220}$ | $-16.67_{197}$ | $0_{118}$ | $0_{125}$ | $1_{19}$ | $0_{183}$ | $0_{220}$ |
| $Ps(S_{3,1} \oplus S_{3,2})$ | $1922_2$ | $66.42_4$ | $-48.55_{123}$ | $0.07_{28}$ | $0.2_{12}$ | $0.67_6$ | $1_3$ | $1.35_6$ |
| $Ps(S_{3,1} \oplus S_{3,2})\ (k = 25)$ | $1922_2$ | $79.45_2$ | $-66.06_{183}$ | $0.08_1$ | $0.23_1$ | $0.57_{27}$ | $0.97_{139}$ | $1.61_2$ |

**Table 9. Case 3 with pseudopaper.** SVA metric scores of the top-10 most cited papers in the network containing $Ps(S_{3,1} \oplus S_{3,2})$) (denoted by ⋆) with scaling factor 25 ($k = 25$; 1988-1993). A total of 208 citing papers made novel co-citation links. SVA score ranges: ΔM:0.0–80.09 | CL:0.0–66.06 | $C_{KL}$:0.0–0.08 | H:0.0–0.23 | $\alpha$:0.0–1.0 | $\beta$:0.0–1.0 | E: 0.0–1.68.

| Citation | ΔM | CL | $C_{KL}$ | H | $\alpha$ | $\beta$ | E | DOI |
|---|---|---|---|---|---|---|---|---|
| 2463 | 0 | -20.43 | 0.01 | 0 | 0.49 | 0.86 | 0.57 | 10.1038/362318a0 |
| ⋆1922 | 79.45[2] | -66.06[183] | **0.08[1]** | **0.23[1]** | 0.57[27] | 0.97[139] | 1.61[2] | N/A |
| 1727 | 0 | -3.55 | 0 | 0 | 0 | 1 | 0.69 | 10.1016/0092-8674(93)90384-3 |
| 1482 | 0 | -11.11 | 0 | 0 | 0.33 | 0 | 0 | 10.1016/0092-8674(93)90376-2 |
| 696 | 0 | -0.67 | 0 | 0 | 1 | 0 | 0 | 10.1111/j.1469-8137.1993.tb03796.x |
| 569 | 0 | 0 | 0 | 0 | 0 | 0 | 0 | 10.1111/1523-1747.ep12359590 |
| 534 | 0 | -2.14 | 0 | 0 | 0 | 1 | 0.69 | 10.1002/j.1460-2075.1993.tb05813.x |
| 518 | 67.81 | -13.7 | 0.01 | 0.02 | 0 | 0.85 | 1.21 | 10.1073/pnas.90.7.2559 |
| 495 | 0 | -5.53 | 0 | 0 | 0 | 1 | 0.69 | 10.1073/pnas.90.7.2812 |
| 495 | 0 | -1.36 | 0 | 0 | 0 | 1 | 0.69 | 10.1128/mmbr.57.2.402-414.1993 |

for pseudopaper $Ps(S_{3,1} \oplus S_{3,2})$ such that it now ranked first according to $C_{KL}$ and $H$. We also found that with $k = 25$ the $\Delta M$ and $E$ scores improved dramatically. We detail the performance of Ohsumi's pseudopaper $Ps(S_{3,1} \oplus S_{3,2})$ ($k = 25$) against the other top cited papers in Table 9. Table 10 shows the step-wise effects of increasing scaling factor $k$ in Case 3 (Ohsumi's case).

Further, we visually demonstrate the positive SVA effect of increasing a network's scaling factor. Contrasting parts **(A)** and **(B)** in Fig 8 suggests that the coverage of pseudopaper $Ps(S_{3,1} \oplus S_{3,2})$'s novel links was expanded significantly when $k$ was set to 25. They reached out to new, far away cluster such as the protein biogenesis cluster (Cluster #5 in part **(B)** of the figure). This could have resulted in the increased centrality divergence and entropy values of the pseudopaper (Table 8). The expanded reach of the novel links is expected because increasing $k$ from 5 to 25 introduced nearly three times more nodes and links into the network. The harmonic mean of the modularity and the weighted mean silhouette of the network also substantially improved from 0.2808 to 0.3767, suggesting an enhanced network quality. Having said that, setting large scaling factors is computationally expensive in CiteSpace and should be employed with care.

The increased scaling factor the network also brought in a new potentially breakthrough paper that was not found previously. A paper by [68], located on the third last row of Table 9 (DOI 10.1073/pnas.90.7.2559), proposed a new synaptic vesicle docking and fusion model. Their model may lead to a better understanding about the molecular mechanisms in neurotransmitter release, which is important to learning and memory in higher organisms. The relatively high $E$ (1.21) and $\Delta M$ (67.81) scores of this paper may signify its boundary-spanning characteristic and high scientific value. In fact, we found that its co-author, Richard H. Scheller, was the recipient of the 1997 NAS Award in Molecular Biology for his work in '*performing*

**Table 10. Effects of $k$ parameter.** The effects of varying scaling factor $k$ on detecting structural variation signals from pseudopaper $Ps(S_{3,1} \oplus S_{3,2})$ are demonstrated for Case 3. The numbers in parentheses indicate to the pseudopaper's ranks by the corresponding metrics.

| $k$ | ΔM | CL | $C_{KL}$ | H | $\alpha$ | $\beta$ | E |
|---|---|---|---|---|---|---|---|
| 5 | 66.42 (5) | -48.55 (123) | 0.07 (28) | 0.20 (12) | 0.67 (6) | 1.00 (3) | 1.35 (6) |
| 10 | 75.83 (3) | -54.01 (157) | 0.08 (19) | 0.24 (10) | 0.8 (15) | 1.00 (9) | 1.52 (3) |
| **15** | **74.20 (7)** | -59.89 (166) | **0.09 (1)** | **0.26 (1)** | **0.86 (13)** | **1.00 (6)** | **1.61 (3)** |
| 20 | 79.28 (3) | -61.90 (173) | 0.08 (2) | **0.25 (1)** | 0.71 (14) | 0.97 (130) | 1.61 (2) |
| **25** | **79.45 (2)** | -66.06 (183) | **0.08 (1)** | **0.23 (1)** | **0.57 (27)** | **0.97 (139)** | **1.61 (2)** |

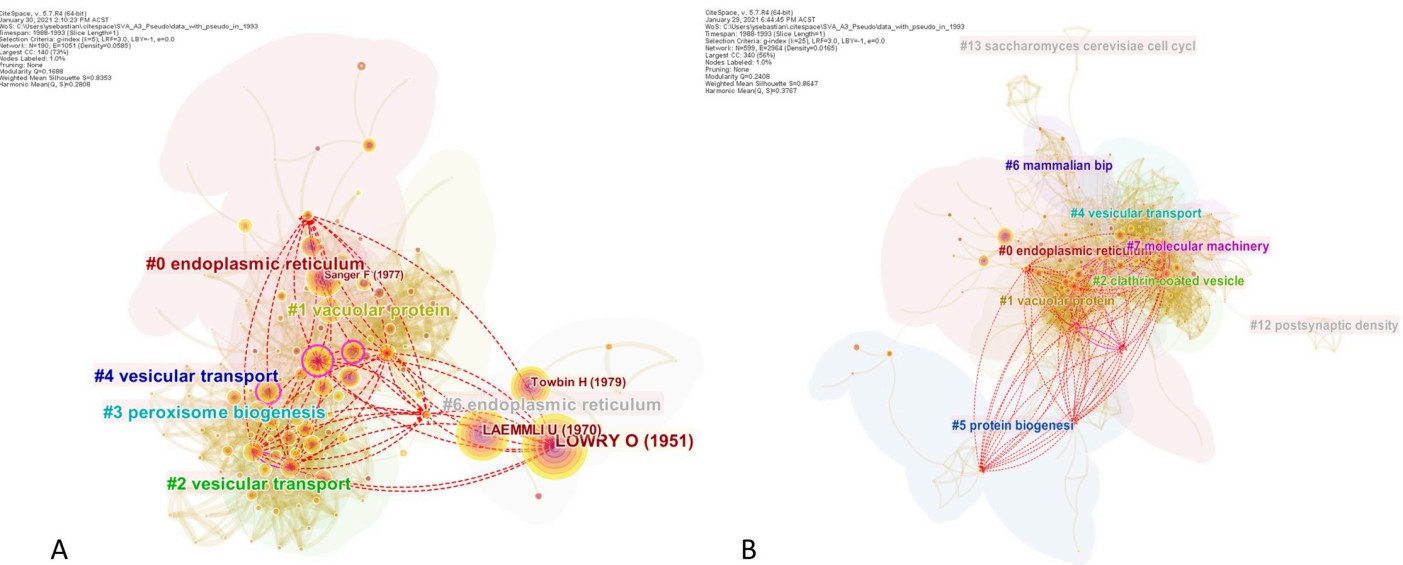

**Fig 8. Visualizing the effects of increased scaling factors.** Increasing the scaling factor of co-citation network in Case 3 (1988–1993) enriches the intellectual representation of the network and broadens the coverage of a pseudopaper's novel links. **(A)** depicts the novel links added by $Ps(S_{3,1} \oplus S_{3,2})$ ($k = 5$). **(B)** shows the novel links added by the same pseudopaper for $k = 25$.

*elegant experiments to resolve the molecular components responsible for controlling neurotransmitter vesicle release and chemical communication within the nervous system*' [69]. Note that several NAS Award recipients went on to receive Nobel Prizes in Physiology or Medicine. This is another example of how SVA could be used to more comprehensively assess and appreciate the important scientific contributions made by other papers that are not necessarily recognized with Nobel Prizes.

Finally, let us make a few notes regarding the selection of $\rho_{S_{2,2}}$ for the purpose of synthesizing $Ps(S_{2,2} \oplus \rho_{S_{2,2}})$ in Case 2. Because the discovery by May-Britt and Edvard Moser was only associated with a single winning paper, in lieu of a second winning paper we decided to incorporate other papers published by these laureates in the same year as that of $S_{2,2,}$. The replacement paper $\rho_{S_{2,2}}$ was limited to only 1 paper to maintain consistency with the pseudopapers for the other two cases. We also searched candidates for $\rho_{S_{2,2}}$ from the same CCE dataset seeded with $S_{2,2}$ to maximize the relevance of the selected paper. Additional heuristics for shortlisting $\rho_{S_{2,2}}$ candidates are: (i) currently listed as candidate winning papers in the source Nobel Prize dataset [33], (ii) co-authored by *both* May-Britt and Edvard Moser, (iii) does not belong to the competing research group led by John O'Keefe, and (iv) demonstrated considerably SVA signals according to $C_{KL}$, $H$, and $E$ (we chose these metrics given their good performance so far). Three candidates emerged from criteria (i)-(iii), namely [70–72]. Using criterion (iv) as a tie-breaker, we selected [72] as $\rho_{S_{2,2}}$.

## Discussions

Our work demonstrates, for the first time, the promising performance of SVA metrics in explaining the boundary-spanning mechanisms behind selected Nobel Prize winning papers. The proposed SVA metrics can capture the impact of award-winning papers in terms of what boundaries they bridged exactly. In contrast to many existing citation-based predictors related

to Nobel Prize which require the accumulation of citations over the years (e.g. [8, 12, 41, 45]), our method can be used to immediately measure the potential impact of scientific papers at the time of their publications. Such benefit is appealing to researchers, publishers, and research policy makers. Our method is transparent, highly reproducible, and requires only minimal adjustments to the standard SVA procedures and parameters in CiteSpace.

Fundamentally, our approach has advantages in terms of *efficiency*, *explanatory power*, and *general practicality*. It is efficient as it is the least data reliant in comparison to other methods. SVA only requires data at the fundamental level of knowledge creation through a paper's cited references without the need for citation data. This means that transformative potentials can be determined much earlier and faster. Given this feature, our method could clearly distinguish the *observations* of intellectual impacts (e.g. citation count) from the *underlying mechanisms* that generate those impacts. This is a benefit that is difficult to achieve with citation-based models because they do not focus on the knowledge creation level. Our approach is also highly explanatory. It provides an extra analytic layer that explains how creative ideas, such as those demonstrated by past Nobel Prize laureates, were constructed by humans in terms of the state of the knowledge they were at. Lastly, in terms of general practicality, SVA's domain agnostic quality means that it could be used to identify boundary-spanning mechanisms that exist in various scientific disciplines. This is practically advantageous in that researchers can replicate such mechanisms in their own scholarly pursuit.

There are a few other advantages afforded by the proposed method. Recent empirical results suggest that the temporality of citations received by a paper is not a reliable predictor of its breakthrough quality, a phenomenon that has at least been observed among Nobel Prize in Physics winning papers [41]. By logical extension, this means that the accumulation of citation counts over a prolonged period of time did not always positively correlate with the perceived impact of a paper according to Nobel Prize award panelists. Consequently, an approach such as SVA that examines the pool of knowledge drawn by a paper rather than the scientific recognition it receives from future papers will be equally, if not more, promising than the citation-based approaches. Also, by not looking at the accumulated citations, our approach is less prone to the Matthew Effect bias in scientific publishing [46]. This is encouraging given that the bias continues to drive citation inequality globally [73]. At last, SVA could potentially be used for early identifications of many 'sleeping beauties', i.e. papers that went unrecognized for years before suddenly becoming highly popular and cited [47]. This is a capability that is difficult to achieve with citation-based approaches as they heavily rely on the availability of citation data.

Specific to the Nobel Prize scenarios, since papers that may never land on a Nobel Prize are far more than those that did, our methodology will be valuable for identifying papers that share the same boundary-spanning properties despite the fact that they may not be awarded a Nobel Prize for one reason or another. This is already demonstrated in our results for Case 2 and Case 3 involving [66, 68], respectively. The method's potentially wide-ranging applicability is important to increase our understanding of the nature of research excellence.

## Nobel Prize and SVA

The Nobel Prize winning papers selected for this study were distinguished by exceptional boundary-spanning characteristics, which are very well captured by the centrality divergence ($C_{KL}$), entropy ($E$), and the harmonic ($H$) metrics. We also show that our artificial pseudopaper approach allows the potential structural variations made by a Nobel Prize discovery to be more amenable to detection by SVA metrics. This technique will be useful in scenarios where the original winning papers tend to emit weak SVA signals.

As for the other seed papers, the initial underperformance of SVA metrics under their individual assessment suggests that the scientific community may have perceived the impact of a seed paper differently, or that history played it out differently. In other words, it cannot be ruled out that some Nobel Prize winning papers may not play the most fundamental scientific role and that one may have overestimated their role in relation to the discovery. There could be other papers which could have better played the role in their place but are not recognized by Li et al's study [33]. Due to the non-trivial nature of nominating and selecting the actual Nobel Prize winners, we also recognize that there could be other selection criteria and other mechanisms of scientific discovery that cannot be fully accounted by the structural variation theory.

### Interpreting SVA metrics

The general behaviours of $\Delta M$, $CL$, and $C_{KL}$ have been discussed elsewhere. For example, the strong betweenness centrality divergence induced by a paper (i.e. high $C_{KL}$ score) was found to be a valuable early sign of its transformative potential at the interdisciplinary levels [5]. Likewise in this study $C_{KL}$ emerged as the clear winner, adding further evidence to its usefulness. The significance of $C_{KL}$ metric suggests that not all novel co-citation links contribute equally to the overall impact of a paper and that the scientific community may tend to favor papers that draw new links in ways that significantly alter the distribution of betweenness centrality scores of nodes in a network. This makes sense given that betweenness centrality characterizes a boundary-spanning mechanism; a node of a high betweenness centrality evidently bridges two bodies of scientific knowledge. We also note that the reported lower performance of $\Delta M$ is consistent with earlier studies [5] and may suggest some limitations of metrics that rely on modularity-based network partitioning.

Metric $E$ and $H$ are relatively newer SVA metrics and their good performance in this study is highly encouraging. $E$ can be considered as a diversity metric that favors the breadth over the depth in terms of the coverage/distribution of cited references over clusters. The metric offers a major advantage in terms of its simplicity and connection to the widely known Shannon's information entropy [30, 74]. Its good performance in relation to the Nobel Prize winning papers may further strengthen the prevailing association between the ability to draw novel links between diverse ideas and the propensity of making lasting scientific impacts [6, 19]. However, the novelty or the uncertainties of some critical ties may be reduced over time as more studies get published. This is a good candidate for further investigations. The promising performance of $H$ suggests the workability of an aggregation-oriented approach to measuring structural variations. It may anticipate a category of breakthrough papers that may not necessarily stand out by a single metric but are exceptional when several metrics are taken together.

Previously, $CL$ was found to correlate well with the future citation counts of research papers [5]. Accordingly, in the current study, its usefulness is limited to seed papers that garnered exceptionally high citation counts. For under-cited influential papers such as Ohsumi's, $CL$ is not a good boundary-spanning indicator. In this case, $CL$ produced negative scores, which is quite an unexpected behavior. We reserve this issue for future investigations.

Finally, we offer a few explanations as to why the artificial pseudopaper technique might have worked. In scenarios where two seed papers were published in proximity to each other, it is possible that the first seed already filled up the intellectual gaps to such extent that there is little left to fill by the second seed. As mentioned earlier, this 'self-eclipse' phenomenon seems to characterize the relationship between $S_{1,3}$ and $S_{1,4}$. Consequently, a pseudopaper could have strengthened the overall impact of both papers, especially when they should be naturally

evaluated in conjunction with each other. This further suggests that research excellence may be more appropriately recognized in a way that transcends the scope of individual publications. The same conceptual consolidation technique may also be beneficial because it overcomes the limitations in scenarios where novel links are so dispersed over a few winning papers that a single paper does not exhibit structural variation strong enough for detection (e.g. Ohsumi's papers in Case 3). It might also help in situations where a Nobel laureate might have drawn crucial novel associations through another less popular paper (e.g. the inclusion of a non-prize winning paper by May-Britt and Edvard Moser provided help for this case).

## Useful SVA strategies

Our experience suggests several new strategies in using SVA with CiteSpace. We observed that increasing the $k$ value quickly enlarges the network, resulting in untenable SVA running time. Therefore, we recommend that anyone interested in using SVA first prioritize the sliding windows before optimizing other configuration parameters such as $k$. The width of the sliding window should be increased gradually (by default is 2) while maintaining a low scaling factor (e.g. 5). This allows a richer structure of the network to emerge while keeping reasonable SVA running time (i.e. within hours instead of days). Besides, we also found that the quality of SVA detection does not always change monotonically with the increase or decrease in sliding window value. So, we suggest that researchers first explore and select the most 'optimal' sliding window under a small, fixed $k$ value. The optimum sliding window size may vary by case. Once a 'promising' window width is determined, use it with increasingly larger $k$ values to obtain better SVA detection. The workability of this approach was previously demonstrated in improving SVA scores of pseudopaper $Ps(S_{3,1} \oplus S_{3,2})$ by setting larger $k = 25$.

## Limitations and future work

We provided cases of boundary-spanning mechanisms that are supported at a higher level of abstraction. We have not addressed in detail the novel conceptual links made by each seed paper at the cluster level. Doing so is necessary to further understand the exact nature of their boundary-spanning mechanisms. The conceptual role played by the seed paper itself is probably the most important one and is unfortunately beyond the expected scope of CiteSpace's SVA. Also, given the inherent limitation associated with most case study-based research, the present findings are only indicative of the boundary-spanning properties of Nobel Prize winners in general.

The Cascading Citation Expansion is a highly systematic method that capitalizes on the conscious and highly selective citing behaviours to gather the most relevant literature base [62]. It is not immediately evident that seeding the CCE process with pre-selected Nobel Prize winning papers could have produced networks that inadvertently amplified the SVA signals of the seed papers. The difficulties in detecting SVA signals of some seeds in the initial part of our experiments may in fact suggest that such biases, if any, have a limited impact on the ensuing analyses. A future work may include empirically testing this assumption by seeding CCE with a non-winning paper that has close topical relevance to the winning paper.

Focusing on the citation contexts and brokerage links between concepts in concept trees from the connected clusters is a promising research direction [30]. Another exciting direction is to include the SVA analyses of papers published by previous Nobel Prize nominees, who competed for the award but did not win. Nomination data are gradually made available through the Nobel Prize Nomination Archive [75].

Future studies should also focus on conducting a larger scale SVA analysis on a higher number of Nobel Prize winning papers to further ascertain the current findings. In this

respect, the algorithmic complexity of SVA implementation by CiteSpace has yet to be systematically studied and reported. However, as mentioned above, our experience suggests that in practice this is likely to be bounded by a combination of several network configurations, such as the scaling factor *k*, size of sliding window, link retaining factor (LRF), and look back years (LBY). Fortunately, these parameters can be controlled and consequently optimized. To extend the current study to all Nobel Prize winning papers researchers should have access to Dimensions' application programming interface (API). This API is a prerequisite to run cascading citation expansion with CiteSpace, whose complexity and optimal configuration (e.g. how far backward or forward should the expansion be) have also yet to be reported [62].

Finally, researchers may consider applying several emerging network analysis paradigms such as network embedding, heterogeneous network representation, and knowledge graphs [76–78]. These techniques could potentially reveal new insights into the boundary-spanning mechanisms of Nobel Prize winning papers in ways that could not be captured by co-citation networks alone.

## Conclusion

The boundary-spanning mechanisms of a select group of Nobel Prize winning papers are demonstrated through a series of experiments with SVA. They are strongly characterized by the ability to draw novel co-citation links from topically-diverse clusters in ways that led to significant alterations of the betweenness centrality distributions in the underlying co-citation network. Through this work, we not only isolated key SVA leading indicators that are representative of the Nobel Prize papers's qualities but also proposed new techniques for improving SVA signal detection. Overall, our findings support the structural variation theory as a promising scientific breakthrough theory.

## Acknowledgments

This paper was written using data obtained in July 2020 from Digital Science's Dimensions platform, available at https://app.dimensions.ai. We thank the anonymous reviewers for their comments.

## Author Contributions

**Conceptualization:** Yakub Sebastian, Chaomei Chen.

**Data curation:** Chaomei Chen.

**Formal analysis:** Yakub Sebastian.

**Investigation:** Yakub Sebastian, Chaomei Chen.

**Methodology:** Yakub Sebastian, Chaomei Chen.

**Resources:** Yakub Sebastian.

**Software:** Chaomei Chen.

**Supervision:** Chaomei Chen.

**Validation:** Yakub Sebastian, Chaomei Chen.

**Visualization:** Yakub Sebastian.

**Writing – original draft:** Yakub Sebastian, Chaomei Chen.

**Writing – review & editing:** Yakub Sebastian, Chaomei Chen.

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
