## [Decision Letter · Decision Letter 0]

10 May 2021

PONE-D-21-07994

The boundary-spanning mechanisms of Nobel Prize winning papers

PLOS ONE

Dear Dr. Sebastian,

Thank you for submitting your manuscript to PLOS ONE. After careful consideration, we feel that it has merit but does not fully meet PLOS ONE’s publication criteria as it currently stands. Therefore, we invite you to submit a revised version of the manuscript that addresses the points raised during the review process.

We look forward to receiving your revised manuscript.

Kind regards,

Yiming Tang, Ph.D.

Academic Editor

PLOS ONE

Journal Requirements:

Reviewers' comments:

Reviewer's Responses to Questions

**Comments to the Author**

1. Is the manuscript technically sound, and do the data support the conclusions?

Reviewer #1: Yes

2. Has the statistical analysis been performed appropriately and rigorously? 

Reviewer #1: Yes

3. Have the authors made all data underlying the findings in their manuscript fully available?

Reviewer #1: No

4. Is the manuscript presented in an intelligible fashion and written in standard English?

Reviewer #1: Yes

5. Review Comments to the Author

Reviewer #1: I enjoyed reading manuscript PONE-D-21-07994 on boundary-spanning mechanisms of Nobel Prize winning papers. Here by looking at the structural impact on co-citation networks, this paper tackles an important question: what are the signals (based on references) that predict breakthrough (Nobel Prize winning) papers? Overall this paper is technically sound and clearly written, and should be accepted after considering the following comments.

1. My only major comment is about the framing of its contribution. There have been many citation-based predictors related to Nobel Prize (on a related note, the authors may also want to cite “Network Signatures of Success: Emulating Expert and Crowd Assessment in Science, Art, and Technology”). The current version distinguishes from itself from these works in timing, i.e. “leading indicators of breakthrough that can be computed at the time of a paper’s publication”. Yet in my opinion, the difference between the two approaches may be even more fundamental – the new analysis presented here is purely based the stock of knowledge, rather than scientific impact or recognition received from future works. Maybe the authors can come up with better words/phrases, but I would recommend thinking a little bit more around the conceptualization.

2. The readers may not be extremely familiar with SVA, so it might be better to provide more explanations for technical terms. For one, the word “cluster” can be ambiguous in network literature – some use it to indicate connected components while others use it to any communities in community detection tasks (which seem to be the case here)

3. I wonder how scalable this approach would be. For example, if we want to run an analysis over all Nobel-prize winning papers, what is the rough estimation of manual efforts one should spend? To be clear I’m not questioning the practical value of this method, but adding some more comments around this would be extremely helpful for future studies (either a highlight of its high scalability or a sentence in limitation should be good).

4. For discussion of future works, a promising direction would be engaging more advanced network analysis tools (e.g. high-order indexes as well as network embeddings) to better understand the predictability of network-based approaches. Again I’m not asking for additional analyses here, but having some discussions around this should make the paper more relevant for a broader range of audience.

6. PLOS authors have the option to publish the peer review history of their article (what does this mean?). If published, this will include your full peer review and any attached files.

Reviewer #1: No

---

## [Author Response · Author response to Decision Letter 0]

22 May 2021

Reviewer’s open comments:

1. My only major comment is about the framing of its contribution. There have been many citation-based predictors related to Nobel Prize (on a related note, the authors may also want to cite “Network Signatures of Success: Emulating Expert and Crowd Assessment in Science, Art, and Technology”). The current version distinguishes from itself from these works in timing, i.e. “leading indicators of breakthrough that can be computed at the time of a paper’s publication”. Yet in my opinion, the difference between the two approaches may be even more fundamental – the new analysis presented here is purely based the stock of knowledge, rather than scientific impact or recognition received from future works. Maybe the authors can come up with better words/phrases, but I would recommend thinking a little bit more around the conceptualization.

Our response:

We thank the reviewer for pointing us to the related work Zakhlebin, I. and Horvát, E.Á., 2017, November. Network signatures of success: Emulating expert and crowd assessment in science, art, and technology. In International Conference on Complex Networks and their Applications (pp. 437-449). Springer. Correspondingly, we added a review of this work in section “Network properties and structural variations”, lines 138 – 146 with a new reference [41] corresponding to it.

As suggested by the reviewer, we further elaborated the conceptualization and impacts of our research contributions on lines 190 – 202 and 696 – 734. We also added new references [46], [47] and [69] in support of the arguments.

2. The readers may not be extremely familiar with SVA, so it might be better to provide more explanations for technical terms. For one, the word “cluster” can be ambiguous in network literature – some use it to indicate connected components while others use it to any communities in community detection tasks (which seem to be the case here).

Our response:

We added two new paragraphs on lines 290 – 306 that provide detailed explanations of two key terminologies underlying SVA: co-citation network and clusters. As correctly observed by the reviewer, the precise meaning of the term `cluster’ in this manuscript carries a similar meaning to `community’ in community detection.

We also included a new reference [57], which provides the interested novice readers with extra insights into the theory and intuition behind co-citation networks. 

3. I wonder how scalable this approach would be. For example, if we want to run an analysis over all Nobel-prize winning papers, what is the rough estimation of manual efforts one should spend? To be clear I’m not questioning the practical value of this method, but adding some more comments around this would be extremely helpful for future studies (either a highlight of its high scalability or a sentence in limitation should be good).

Our response:

We added 1 paragraph under ‘Limitation’ section on lines 846 – 857 on the scalability of our approach and current limitations.

4. For discussion of future works, a promising direction would be engaging more advanced network analysis tools (e.g. high-order indexes as well as network embeddings) to better understand the predictability of network-based approaches. Again I’m not asking for additional analyses here, but having some discussions around this should make the paper more relevant for a broader range of audience.

Our response:

We added 1 paragraph under ‘Limitation’ section on lines 858 – 862, suggesting that recent network analysis paradigms (such as network embedding, heterogeneous network representation, and knowledge graph learning) may provide possible alternatives for measuring boundary-spanning qualities of Nobel Prize winning papers. New references [71 – 73] were added in support of these.

 

Dataset availability:

[3.] Have the authors made all data underlying the findings in their manuscript fully available?

Reviewer #1: No

Our response: 

The datasets we generated and analyzed for this study are freely accessible from Figshare at https://figshare.com/s/5e0e279c51f6de9df947 . Further information about the dataset provision and access is provided in the S1 file (Supporting Information).

Acknowledgment:

We added a thank you note to the anonymous reviewer.

---

## [Decision Letter · Decision Letter 1]

6 Jul 2021

The boundary-spanning mechanisms of Nobel Prize winning papers

PONE-D-21-07994R1

Dear Dr. Sebastian,

We’re pleased to inform you that your manuscript has been judged scientifically suitable for publication and will be formally accepted for publication once it meets all outstanding technical requirements.

Kind regards,

Yiming Tang, Ph.D.

Academic Editor

PLOS ONE

Additional Editor Comments (optional):

Reviewers' comments:

Reviewer's Responses to Questions

**Comments to the Author**

1. If the authors have adequately addressed your comments raised in a previous round of review and you feel that this manuscript is now acceptable for publication, you may indicate that here to bypass the “Comments to the Author” section, enter your conflict of interest statement in the “Confidential to Editor” section, and submit your "Accept" recommendation.

Reviewer #1: All comments have been addressed

Reviewer #2: All comments have been addressed

2. Is the manuscript technically sound, and do the data support the conclusions?

Reviewer #1: Yes

Reviewer #2: Yes

3. Has the statistical analysis been performed appropriately and rigorously? 

Reviewer #1: Yes

Reviewer #2: Yes

4. Have the authors made all data underlying the findings in their manuscript fully available?

Reviewer #1: Yes

Reviewer #2: Yes

5. Is the manuscript presented in an intelligible fashion and written in standard English?

Reviewer #1: Yes

Reviewer #2: Yes

6. Review Comments to the Author

Reviewer #1: I appreciate the authors' efforts in revising this paper and would like to recommend publication of its current version. Thanks for this interesting paper.

Reviewer #2: The paper is technically sound and the method is good. More detailed explanations of previous works may be included.

7. PLOS authors have the option to publish the peer review history of their article (what does this mean?). If published, this will include your full peer review and any attached files.

Reviewer #1: No

Reviewer #2: No

---

## [Editor Report · Acceptance letter]

21 Jul 2021

PONE-D-21-07994R1 

The boundary-spanning mechanisms of Nobel Prize winning papers 

Dear Dr. Sebastian:

I'm pleased to inform you that your manuscript has been deemed suitable for publication in PLOS ONE. Congratulations! Your manuscript is now with our production department. 

Kind regards, 

on behalf of

Professor Yiming Tang 

Academic Editor

PLOS ONE